# Single cell and spatial transcriptomics highlight the interaction of club-like cells with immunosuppressive myeloid cells in prostate cancer

Prostate cancer treatment resistance is a significant challenge facing the field. Genomic and transcriptomic profiling have partially elucidated the mechanisms through which cancer cells escape treatment, but their relation toward the tumor microenvironment (TME) remains elusive. Here we present a comprehensive transcriptomic landscape of the prostate TME at multiple points in the standard treatment timeline employing single-cell RNA-sequencing and spatial transcriptomics data from 120 patients. We identify club-like cells as a key epithelial cell subtype that acts as an interface between the prostate and the immune system. Tissue areas enriched with club-like cells have depleted androgen signaling and upregulated expression of luminal progenitor cell markers. Club-like cells display a senescence-associated secretory phenotype and their presence is linked to increased polymorphonuclear myeloid-derived suppressor cell (PMN-MDSC) activity. Our results indicate that club-like cells are associated with myeloid inflammation previously linked to androgen deprivation therapy resistance, providing a rationale for their therapeutic targeting.

Approximately one in eight men develop prostate cancer (PCa) during their lifetime. A favorable prognosis for localized tumors is contrasted by the high mortality rate and challenging treatment landscape of metastatic disease[1]. Genomics and transcriptomics data from clinical samples have revealed the various aberrations underlying primary[2,3] and treatment-resistant tumors[4,5]. Single-cell RNA-sequencing (scRNA-seq) studies have elucidated the heterogeneity inside these tumors, demonstrating that bulk-level investigations fail to address their whole complexity[6–9]. While powerful, scRNA-seq is hampered by a lack of spatial information making it a suboptimal tool for observing interactions in the tumor microenvironment (TME). A handful of studies have investigated the prostate TME in a spatial context, reporting insights such as transcriptomic differences between Gleason grades[10], malignant cells in histologically benign areas[11], and immunosuppressive regulatory T-cell activity in

coordination with myeloid cells[12], but have included only few patients. The tumor cells' interaction with the TME is a question of high interest, as mounting evidence suggests that the TME actively participates in the progression, treatment response, and metastasis of various tumors[13–15].

The scRNA-seq-based characterization of PCa has led to an appreciation of epithelial cell type heterogeneity, identifying four specialized subtypes: luminal, basal, hillock, and club cells[16,17]. Basal and luminal epithelium are well-described parts of the normal prostate glandular structure. The luminal epithelium is regarded as the cellular origin of prostate adenocarcinomas[18,19]. Hillock cells are a source of discrepancy as they have been claimed to be found[9,12,16,20] or to be undiscernible[17], depending on the study. In some cases, hillock and club cells have been described together as *intermediate* epithelial cells, reflecting the similarity of these cell populations[7,21].

e-mail: alfonso.urbanucci@tuni.fi; matti.nykter@tuni.fi

The role of club cells in the prostate is poorly understood. Club cells were originally described in the lung and characterized for their expression of uteroglobin *SCGB1A1*[16,22]. Initial observations noted that club cells reside in the urethral epithelium and proximal prostate ducts[20,23]. In contrast, more recent work has highlighted their presence in the peripheral zone and inflamed regions of the prostate[24]. Intriguingly, human club cells have a transcriptomic profile similar to the castration resistant luminal progenitor phenotype in mice[25–27], raising questions regarding their function in response to androgen deprivation therapy (ADT) in humans.

Here we investigate the cellular composition of the prostate TME and its response to androgen deprivation. We generate spatial transcriptomics (ST) data from benign prostatic hyperplasia, treatment-naïve primary PCa, neoadjuvant-treated PCa, and castration-resistant PCa, representing a cross-section of cancer and benign prostate disease states together with TME from tissue adjacent to diseased areas. We augment these ST data with publicly available scRNA-seq data, analyzing high-resolution transcriptomics data from 120 patient samples in total. Our study provides an exhaustive map of the prostate TME, ranging from benign tissue to treatment-naïve tumors and advanced, treatment-resistant disease.

## Results

We used the Visium Spatial Transcriptomics assay to generate data from 80 fresh frozen tissue sections from 56 prostatectomy samples. Processed samples contained 1835 tissue-covered spots, 3275 median genes per spot, and 9549 median UMIs per spot on average (Supplementary Data S1). These samples were divided into two groups, constituting a discovery (48 sections from 48 patients) and a validation cohort (32 sections from 8 patients, Fig. 1a). We used the discovery cohort, consisting of treatment-naïve benign prostatic hyperplasia (BPH) and prostate cancer (TRNA), neoadjuvant-treated prostate cancer (NEADT), and castration-resistant prostate cancer tumors (CRPC) to develop a computational pipeline for unsupervised exploration of the prostate TME. We used the validation cohort of treatment-naïve primary tumors to cross-compare results and test hypotheses arising from the discovery cohort.

### Sample-independent deconvolution of spatial transcriptomics data with a single cell-derived cell state reference

The ST assay captures polyadenylated mRNAs onto 55 μm sized uniquely barcoded spots. The number of individual cells and cell types underlying this measured expression varies, presenting a major challenge in the analysis of these data[28,29]. A commonly adopted analysis strategy uses scRNA-seq-derived transcriptomics profiles to infer the most probable cell type compositions in each ST spot[30]. These inferred cell type counts can further be used to divide tissue structures into distinct, biologically meaningful regions[31].

We re-analyzed previously published scRNA-seq datasets from normal prostate tissue[12,32], low- and high-grade primary tumors[7,9,12,17], and both locally recurrent and metastatic castration-resistant prostate cancer tumors[6,8] to assemble a highly diverse cell-state reference likely to capture a similar heterogeneous cell type distribution as the discovery cohort. Using unsupervised clustering and non-negative matrix factorization (NMF) to define tissue compositions across samples (Supplementary Methods), we identified 26 cell states in 223,881 cells across 98 samples and 64 patients (Supplementary Data S2). We used this cell state reference to model the most probable cell state composition in spatial locations of our discovery cohort[31].

Notable redundancy was still present in the inferred cell state counts, as thirteen cell states with the lowest inferred abundance accounted for just 6.7% of the total inferred cell counts (Supplementary Data S3). To address this, we categorized individual spots into eight categories based on their majority cell state contributors (Methods).

This resulted in commonly co-localized cell populations, hereon called single-cell mapping-derived regions (SCM-regions, Fig. 1b).

### Single-cell mapping-derived regions capture well-established biology

We identified eight SCM regions across the discovery cohort. For each sample, we compared the expression between regions to find differentially expressed genes (One-vs-rest two-sided Wilcoxon rank-sum test, Supplementary Data S4). We then determined genes that were enriched among the overexpressed genes in each region ($p_{adj}$ <0.05, one-sided Fisher's exact test), terming these *region-specific markers* (Supplementary Data S5).

We annotated the SCM regions according to their region-specific markers (Fig. 1c). The *Tumor* region overexpressed known PCa-specific markers such as *AMACR*[33], *PCA3*[34], and *PCAT14*[35]. Canonical luminal cell markers *MSMB, ACPP*, and *CD38* were markers for the *Luminal* region, while *KRT5, KRT15*, and *TP63* were overexpressed in the *Basal* region. The *Immune* region overexpressed chemokine receptors (*CXCR3, CXCR4*), T cell- (*TRBC1, TRBC2, CD3D*), and B cell-specific (*CD79A, CD22*) genes, as well as myeloid-lineage marker genes (*LYZ, HLA-DRA, CD74, C1QA, C1QB*). Stromal regions *Endothelium* (*VWF, EPAS1, EMP1*), *Fibroblast* (*DCN, FBLN1, LUM*), and *Muscle* (*ACTA2, TAGLN, MYL9*) were similarly annotated. One SCM region overexpressed recently proposed markers for club cells[17,24] *MMP7, PIGR, CP*, and *LTF*, and was consequently named the *Club* region.

Tumor, Luminal, Basal, Club, Immune, Fibroblast, and Muscle regions were reproduced in the validation cohort following an identical computational workflow. We identified an additional region undetected in the discovery cohort, expressing *FOS, JUN*, and *EGR1* which we consequently named the *Stressed luminal* region. Both *Endothelium* and the *Stressed luminal* regions were the smallest in their cohorts representing 1.8% and 2.2% of all spots, respectively, and may represent differences due to the nature of the composition of the two cohorts. Each spot's neighborhood was enriched for spots of the same region in both cohorts (Supplementary Fig. S1).

SCM regions agreed with the pathologists' assessment of sample histology (Fig. 2a). To quantitively assess SCM region accuracy, we categorized samples in the validation cohort into *Low, Mid*, and *High cancer %* groups according to the fraction of spots classified as cancer by the pathologists (Supplementary Fig. S1). *Low cancer %* samples had a lower fraction of Tumor region spots than *Mid cancer %* ($p = 1.2×10^{-4}$, two-sided Wilcoxon rank-sum test) and *High cancer %* samples ($p = 1.1×10^{-4}$). *High cancer %* samples had lower Basal region fraction than *Mid cancer %* ($p = 3.5×10^{-3}$) or *Low cancer %* ($p = 1.4×10^{-4}$) samples. The fraction of Luminal ($p = 1.6×10^{-2}$, Kruskal-Wallis test) and Muscle ($p = 2.9×10^{-4}$) regions also varied across sampling locations while the fraction of Club, Immune, Stressed, and Fibroblast did not.

In the validation cohort, spots annotated as cancer with ISUP grade group grading were enriched for the Tumor region ($p < 2.2×10^{-16}$, two-sided Fisher's exact test). 67% of the spots deemed normal by the pathologists' were Luminal, Basal, or Club region spots (Fig. 2b). Of the spots annotated as lymphocytes by the pathologists, 90% were labeled as the Immune region. Similar concordance of histopathology annotation and SCM regions was observed in the discovery cohort (Supplementary Fig. S2).

### Androgen deprivation promotes basal and club-like epithelial phenotypes

To investigate the effect of ADT on gene expression, we analyzed the epithelial SCM regions in pre- and post-treatment samples. We observed decreased expression of androgen receptor-regulated (AR-regulated) genes in the Tumor and Luminal regions following treatment (Fig. 2c). These genes were expressed at low levels in the Basal and Club regions in pre- and post-treatment samples. Conversely,

canonical basal cell markers *KRT5*, *KRT15*, and *TP63* were highly expressed in the Basal and Club regions both pre- and post-treatment.

To test how treatment affected gene expression in the Club region specifically, we calculated differentially expressed genes between the Club regions of BPH, TRNA, NEADT, and CRPC sample categories (Two-sided Wilcoxon rank-sum test, Supplementary Data S6). The expression of AR-regulated genes *ABCC4*, *KLK3*, *MAF*, *NKX3-1*, and *PMEPA1* was lower in NEADT than TRNA samples ($log_2$ fold-change $\leq -1$, $p_{adj} < 0.05$, two-sided Wilcoxon rank-sum test). The expression of canonical club cell marker *LTF was* likewise down-regulated, while the expression of *MMP7*, *PIGR*, *SCGB1A1*, and *SCGB3A1* was unaffected. Mouse luminal progenitor marker genes[36]

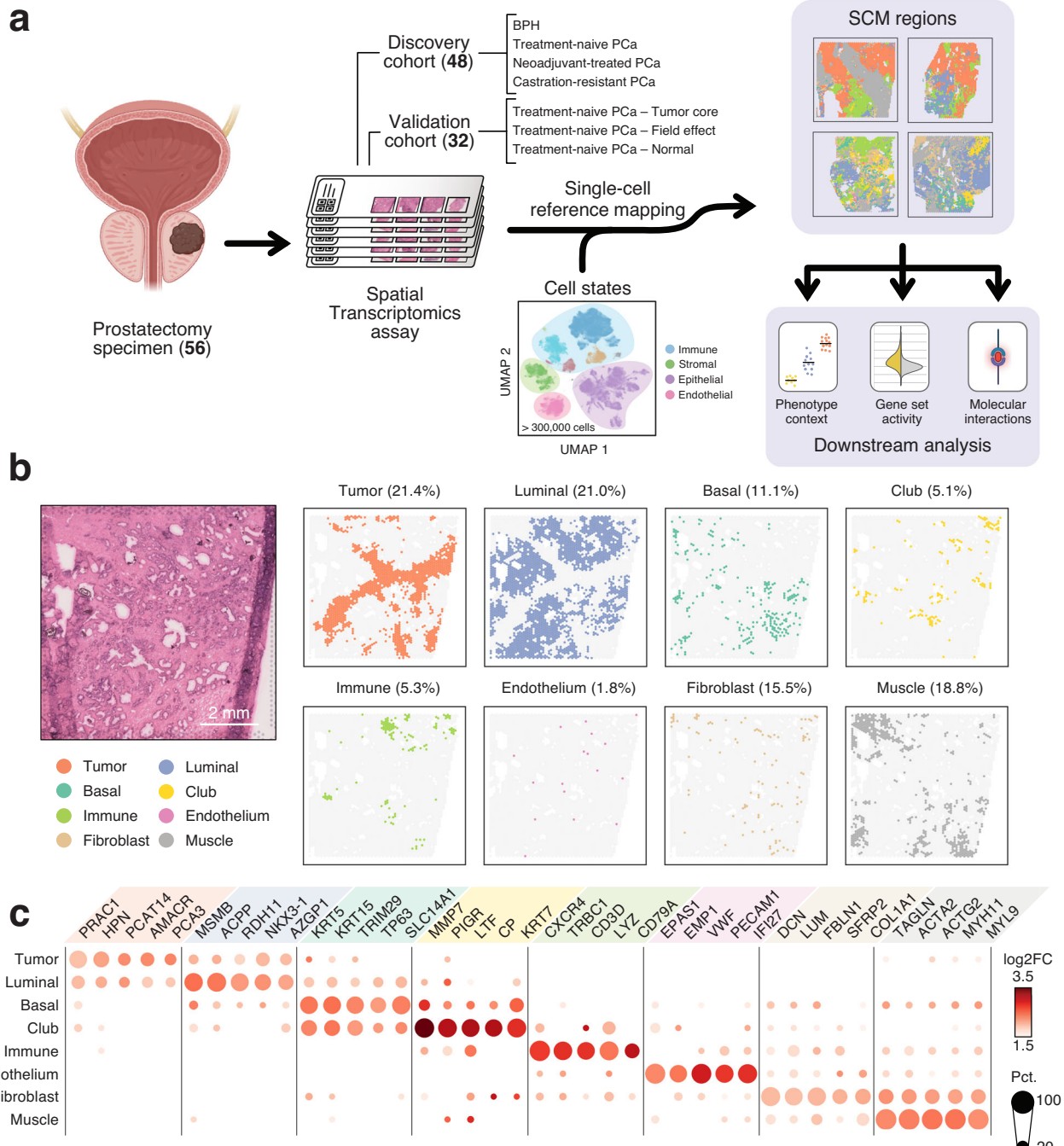

**Fig. 1 | Integration of scRNA-seq and ST data reveals the organizational patterns of the prostate TME. a** Sample collection and analysis pipeline overview. Brackets indicate the number of samples in each category. The cell state reference was assembled from previously published single-cell RNA-sequencing data[6–9,12,17,32]. Created in BioRender. Nykter, M. (2023) https://BioRender.com/n30i289 **b)** An untreated primary tumor sample with eight SCM regions shown separately. Percentages represent the share of spots across the discovery cohort. For details on how the SCM regions were calculated see Methods. Scale bar is 2 mm. **c** Expression of cell type gene markers across the discovery cohort. Each region was tested for differentially expressed genes individually in all samples. Dot size represents the percentage of samples in which the gene was overexpressed (Wilcoxon rank-sum test $p_{adj} < 0.05$ & $log_2$ fold change $\geq 1$). Dot color represents the average log-fold change. Region-specific marker, and their enrichment test $p_{adj}$ (one-sided Fisher's exact test), and region-specific marker status are indicated in Supplementary Data S5. The number of region-specific markers for each region: Tumor (569), Luminal (1,776), Basal (46), Club (452), Immune (594), Endothelium (139), Fibroblast (294), Muscle (280). SCM regions single-cell mapping-based regions.

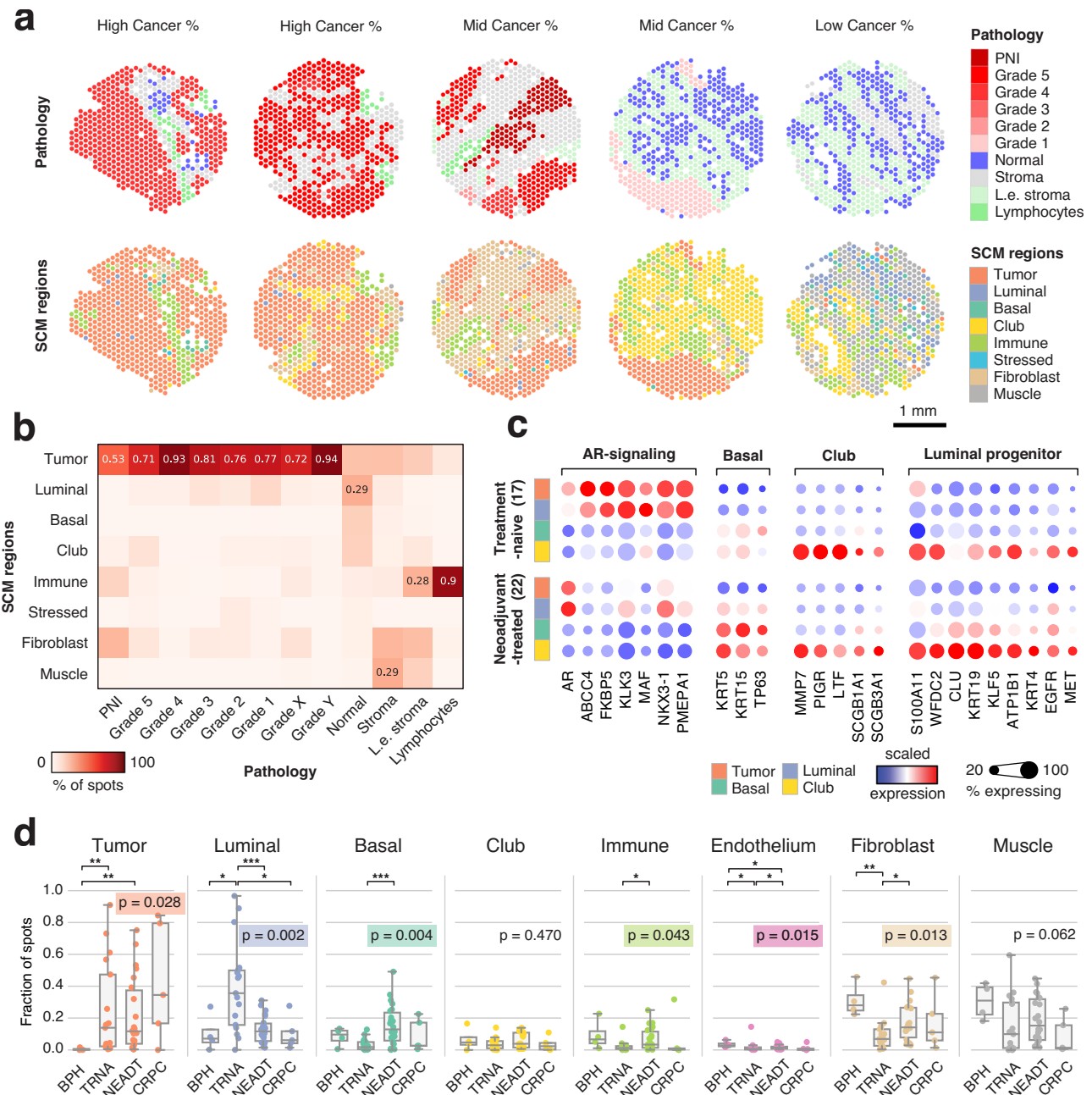

**Fig. 2 | SCM regions match histopathological features and reflect treatment-dependent shifts in gene expression. a** Histopathology classes and SCM regions of five representative samples from the validation cohort. Grade X and Grade Y are cancer spots with indecisive grading (see Methods). PNI: perineural invasion; L.e. stroma: lymphocyte-enriched stroma. Scale bar is 1 mm. **b** Percentage of SCM regions in each histopathology category in the validation cohort. Values are shown for each region with the highest percentage. **c** Scaled gene expression in SCM regions grouped by ADT exposure. **d** Sample-specific SCM region fractions divided according to sample category: BPH: Benign prostatic hyperplasia ($n = 4$); TRNA: Treatment-naïve prostate cancer ($n = 17$); NEADT: Neoadjuvant-treated prostate cancer ($n = 22$); CRPC: Castration-resistant prostate cancer ($n = 5$). A Kruskal-Wallis test p-value is displayed for each category, while asterisks indicate two-sided Wilcoxon rank-sum test significance level; $*p < 0.05$; $**p < 0.01$; $***p < 0.001$. Boxes span the interquartile range (IQR) and whiskers extend to points that lie within 1.5 IQRs of the lower and upper quartile. Center line is drawn at the median.

*S100A11, WFDC2, KRT19, KLF5, ATP1B1, KRT4*, and *MET* were region-specific markers for the Club region, and their expression was similarly unperturbed by treatment.

Compared to BPH, NEADT, and CRPC, the most significantly upregulated DEGs in the TRNA Club region included AR-regulated luminal cell markers such as *KLK3, KLK2, PMEPA1, MSMB, and ACPP* (Supplementary Fig. S3). Club-like cells in tumor tissue have been reported to have increased AR-signaling activity compared to healthy prostate tissue[17]. The upregulation of activating transcription factors

*FOS* and *JUN* was also detected in TRNA samples when compared to BPH samples, the expression of which has been previously linked to epithelial cell stress response in tumor progression[17].

The percentage of Tumor, Luminal, Basal, Immune, Endothelium, and Fibroblast spots varied across the treatment categories (Fig. 2d). TRNA ($p = 7.2×10^{-3}$, two-sided Wilcoxon rank-sum test) and NEADT ($p = 4.5×10^{-3}$) samples had a higher percentage of tumor spots than BPH samples. NEADT samples contained a lower percentage of Luminal region than TRNA samples ($p = 9.2×10^{-4}$), while a higher proportion

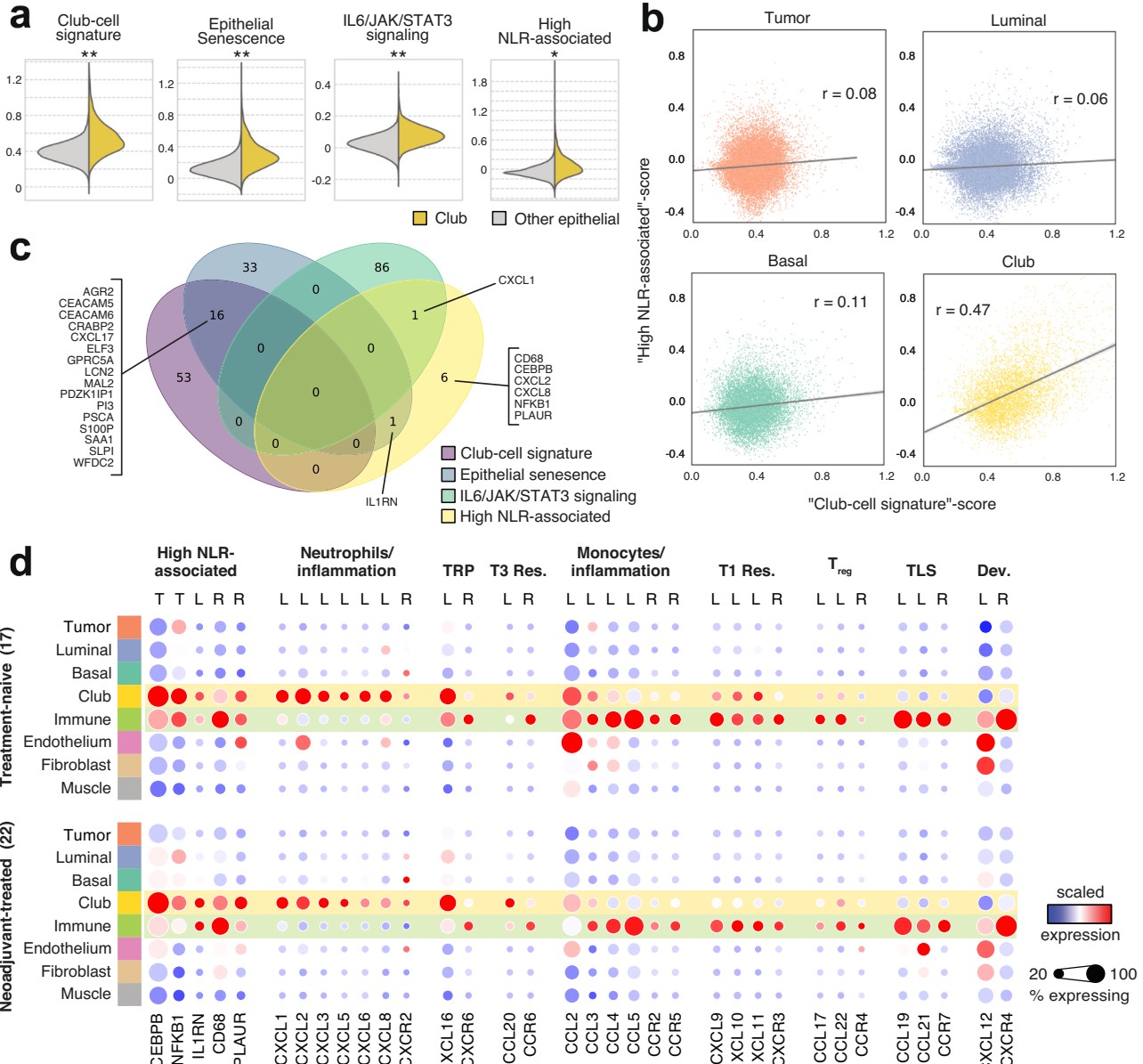

**Fig. 3 | The Club region has increased expression of genes linked to a senescence-associated secretory phenotype. a** Spotwise gene set activity scores in the Club region (*n* = 5595) compared to other epithelial regions (*n* = 59,198). Asterisks indicate two-sided independent samples t-test p-value (*p* < 0.05) and effect size: *Other 70th percentile <Club region mean; **Other 80th percentile <Club region mean; ***Other 90th percentile <Club region mean **b** Gene set score correlation for club-cell signature and high neutrophil-to-lymphocyte ratio-

associated (NLR) signature in epithelial regions. **c** A Venn diagram of gene sets shown in (**a**). **d** Scaled gene expression in SCM regions divided by sample category. Genes are grouped according to immunological function. TRP: tissue-resident program, T3 Res: Type 3 immune response, T1 Res: Type 1 immune response, T_reg: regulatory T cells, TLS: Tertiary lymphoid structure, Dev.: Developmental program, T: transcription factor, L: chemokine ligand, R: chemokine receptor.

of Basal region (*p* = 4.4×10⁻⁴). These results indicate an increase in the fraction of basal phenotype epithelial cells in response to ADT.

**The Club region is tied to inflammation and senescence-associated secretory phenotype**

To elucidate the role of club-like cells in the prostate TME, we calculated spot-level gene set activity scores on prostate-specific[16,37,38], pan-cancer associated[39,40], hallmark signaling[41,42], and immune activity-related gene sets[12,43–46] and compared the scores across epithelial cell regions (Supplementary Data S7). *Club cell*[16], *epithelial senescence* (*EpiSen*)[40], *IL6/JAK/STAT3 signaling*, and *high neutrophil-to-lymphocyte ratio associated* (*high NLR-associated*) gene signatures scored higher in the Club region (*p < 0.05, two-sided independent samples t-test and quantile thresholds*, Fig. 3a). Genes related to androgen response

scored lower in the Club region, while the mouse *luminal progenitor*[36], *lung club cell*[47], *stem cell-like CRPC*[38], and *PROSGenesis*[37] signatures scored higher (Supplementary Fig. S4). Multiple inflammation-related hallmark gene signatures scored higher in the Club region, including *inflammatory response*, *interferon gamma signaling*, *interferon alpha signaling*, and *KRAS signaling UP* (Supplementary Fig. S4).

The expression of *high NLR-associated* genes in CRPC primary tumors correlates with the NLR measured from peripheral blood[45]. High blood NLR is associated with shorter survival and treatment resistance[45,48,49]. We detected a four-fold higher correlation between the club cell and high NLR-associated signature scores in the Club region compared to the other epithelial regions (Fig. 3b), despite no overlap between the two signatures (Fig. 3c). We consolidated gene set activity scores with an enrichment analysis between region-specific

gene markers and these gene sets (Supplementary Fig. S5, Supplementary Data S8). Genes of the *high NLR-associated* signature were overrepresented among Club region markers ($p_{adj} = 1.5 \times 10^{-3}$, one-sided Fisher's exact test). One of the genes in *high NLR-associated* signature, *NFKB1*, codes for nuclear factor κB (NF-κB). *TNF-α-signaling via NF-κB* was among the most significantly enriched gene sets for the Club region ($p_{adj} = 2.7 \times 10^{-26}$). The p53 pathway ($p_{adj} = 2.3 \times 10^{-7}$) hallmark gene set was uniquely enriched in the Club region, in line with its putative senescent nature[50].

Club region-specific markers were enriched for genes that encode for proteins whose secretion is increased in the senescence-associated secretory phenotype[51] (SASP) suggesting that club-like cells engage in similar secretory activity ($p_{adj} = 7.2 \times 10^{-4}$). SASP is typified by the secretion of chemokines, small molecules responsible for inducing chemotaxis of immune cells that have a significant role in TME organization[52,53]. We examined the expression of key chemokines across all SCM regions in a pre- and post-treatment setting (Fig. 3d). Neutrophil chemotaxis-inducing chemokines *CXCL1* ($p_{adj} = 1.5 \times 10^{-13}$, one-sided Fisher's exact test)*, CXCL2* ($p_{adj} = 7.4 \times 10^{-13}$), and *CXCL8* ($p_{adj} = 1.7 \times 10^{-5}$) were region-specific markers for the Club region, along with *CXCL16* ($p_{adj} = 1.5 \times 10^{-13}$) and *CCL20* ($p_{adj} = 2.0 \times 10^{-4}$). Canonical receptors of these chemokines[53] were region-specific markers in the Immune region: *CXCR6* ($p_{adj} = 2.7 \times 10^{-2}$) for *CXCL16* and *CCR6* ($p_{adj} = 4.1 \times 10^{-4}$) for *CCL20*. While the sensitivity of ST was not sufficient to detect significant levels of neutrophil chemotactic receptor *CXCR2* expression, the elevated expression of its canonical ligands *CXCL1, CXCL2,* and *CXCL8* suggests that club-like cells partake in their recruitment to the TME.

## Areas proximal to the Club region have increased polymorphonuclear myeloid-derived suppressor cell activity

Overrepresentation analysis of Gene Ontology: Biological Process (GO:BP) terms revealed enrichment of myeloid cell chemotaxis-associated gene sets in the Club region (Fig. 4a). Lymphoid and myeloid immune cell activation-related processes were enriched in the Immune region. Two gene sets indicating prostate-specific myeloid-derived suppressor cell (MDSC) activity derived from mice[43] ($p = 4.3 \times 10^{-7}$) and human[12] ($p = 2.0 \times 10^{-6}$) were also enriched in the Immune region (Supplementary Data S8). A CRISPR-Cas9 validated signature of polymorphonuclear MDSC (PMN-MDSC) immunosuppression[46] was enriched in the Immune region ($p = 1.9 \times 10^{-2}$), indicating an accumulation of immunosuppressive PMN-MDSCs in the prostate TME.

The fraction of Club region spots in a sample correlated positively with its PMN-MDSC activity score calculated from non-Club region spots in both cohorts (Fig. 4b). Tumor region spots had a higher PMN-MDSC activity score when proximal to Club region spots (Fig. 4c). Similar results were acquired for three other MDSC signatures (Supplementary Fig. S6). One MDSC activity signature scored higher in the recurrent samples ($n = 20$) than in non-recurrent samples ($n = 12$) of the validation cohort ($p = 1.2 \times 10^{-4}$, two-sided Wilcoxon rank-sum test). PMN-MDSC activity scores didn't differ between the TRNA and NEADT groups ($p \geq 0.05$, two-sided independent samples t-test, Fig. 4d).

To validate the observed proximity between club-like cell prevalence and PMN-MDSC infiltration, we performed multiplex immunohistochemistry staining of 16 primary untreated prostate tumor tissue sections using antibodies against previously validated markers for these cell types (Methods). From the resulting staining images, we selected 101 approximately 3 mm² regions of interest that were either club-like negative (absent *LTF* staining, $n = 54$) or club-like positive (moderate to strong *LTF* staining, $n = 47$) (Fig. 4e, Fig. 4f, Supplementary Fig. S7). We then trained a random trees cell classifier for seven cell categories, including PanCK⁺LTF⁺/PIGR⁺ club-like cells[24] and CD45⁺CD66b⁺CD11b⁺CXCR2⁺ PMN-MDSCs[45] (Fig. 4g, Methods).

Across all ROIs, the number of club-like-classified cells correlated positively with the number of PMN-MDSC-classified cells (Fig. 4h). A higher proportion of PMN-MDSCs was present in club-like positive ROIs than in club-like negative ROIs ($p = 1.7 \times 10^{-8}$, two-sided Wilcoxon rank-sum test) (Fig. 4i). No difference in the total number of detected cells between club-like positive and negative ROIs was observed ($p = 0.23$, two-sided Wilcoxon rank-sum test), while a greater number of club-like cells was present in the club-like positive ROIs ($p = 3.0 \times 10^{-15}$) (Supplementary Fig. S7). Taken together these results demonstrate that the presence of club-like cells is strongly associated with PMN-MDSC infiltration and immunosuppressive activity in the prostate TME.

## Club-derived ligand-receptor signaling activity is specific to the interacting region

To elucidate the specific molecular interactions occurring between the Club and other regions, we performed ligand-receptor binding analysis in the Club region interfaces (Supplementary Fig. S8, Supplementary Data S9). Briefly, we surveyed spots of each region that were proximal to Club region spots and tested for interactions between these two sets (Methods). We filtered the interactions to those that had a ligand (from Club to other regions) or receptor (from other regions to Club) overexpressed in the Club region. We then tested for region interface-specific enrichment by comparing the number of interfaces where a specific ligand-receptor pair was active.

This analysis identified 142 unique ligand-receptor interactions occurring at the Club region interface, 59 (41.5%) of which were enriched at specific interfaces ($p_{adj} < 0.05$, one-sided Fisher's exact test). *Atypical chemokine receptor 1 (ACKR1)* was the most common target for neutrophil chemotaxis-inducing chemokines *CXCL1, CXCL2, CXCL5, CXCL6,* and CXCL*8* (Supplementary Fig. S8). The *CXCL2* to *ACKR1* interaction was enriched at the Club-Fibroblast interface ($p_{adj} = 2.3 \times 10^{-2}$). *Dipeptidyl peptidase-4 (DPP4)* was a unique target of *CXCL2* and enriched at the Club-Luminal interface ($p_{adj} = 5.7 \times 10^{8}$, Supplementary Fig. S8). *CCL20* and *CCR6* formed an interaction that was enriched in the Club-Immune interface ($p_{adj} = 2.2 \times 10^{-3}$).

The Club region overexpressed receptors with known oncogenic properties *(MET, EPHA2)*[54,55], inflammatory activities (*CEACAM1, PTGS2*)[56,57], and adhesive function (*ITGA3, ITGA6*) (Supplementary Fig. S8). The Fibroblast-Club interface was enriched for *MET* interaction with *decorin* (*DCN*, $p_{adj} = 1.0 \times 10^{-6}$) and *hepatocyte growth factor* (*HGF*, $p_{adj} = 5.6 \times 10^{-5}$). *EPHA2* was the target of multiple *ephrin*-family ligands, of which *EFNA1* and *EFNA5* signaling interactions were enriched at the Luminal ($p_{adj} = 1.0 \times 10^{-6}$) and Muscle ($p_{adj} = 2.4 \times 10^{-2}$) interfaces, respectively. *TP53* and *PTGS2* interaction was enriched at the Tumor-Club interface ($p_{adj} = 3.0 \times 10^{-2}$). An interaction between Immune region-expressed *CCL5* and Club region-expressed *SDC1* was the single most enriched interaction between these two regions ($p_{adj} = 5.9 \times 10^{-5}$). These results elucidate the region-specific signaling relied upon by the club-like cells.

## Club-like senescence is associated with immunosuppressive PMN-MDSC activity in primary and metastatic tumors

We further examined the connection between the Club region SASP and PMN-MDSC activity. We detected a significant overlap between the *EpiSen* signature and the Club region markers ($p = 3.2 \times 10^{-22}$, one-sided Fisher's exact test), as well as the *EpiSen* signature and PMN-MDSC activity signature ($p = 2.1 \times 10^{-5}$, Fig. 5a). We termed the 26-gene overlap between *EpiSen* and Club region markers *club-like senescence*.

Of the genes overlapping between these three signatures, we found eleven genes to be expressed in the neutrophil and monocyte populations of metastatic CRPC (mCRPC) scRNA-seq data[58], suggesting that similar senescence-driven processes could be taking place in the metastatic tumor environment (Fig. 5b). To investigate, we generated ST data from four metastatic prostate cancer tumors collected

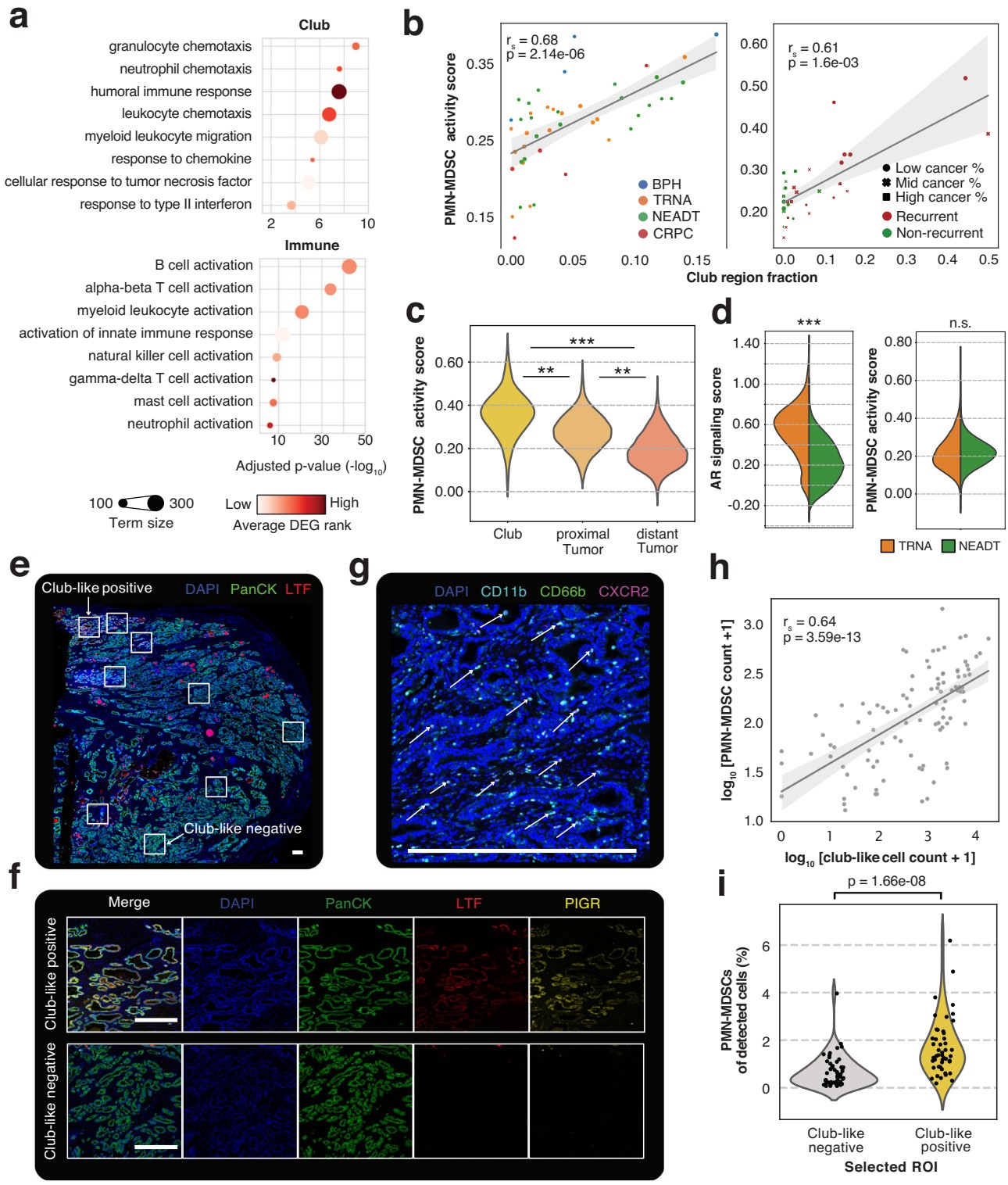

from a pelvic lymph node, liver, pericardial region, and the subdural region, each from a different patient (Methods, Supplementary Data S10). We determined expression-based clusters for each sample separately to find spots with similar expression and underlying cell populations (Fig. 5c, Supplementary Fig. S9). We then performed enrichment analysis and calculated gene set activity scores to determine whether any of these clusters corresponded to the club-like cell population identified in primary tumors.

Clusters *Met A1* (liver metastasis) and *Met B5* (subdural metastasis) displayed elevated scores for the Club region markers and the *club-like*

*senescence* signature (Fig. 5d). These tissue sites overexpressed genes (log_2 fold-change ≥ 1, $p_{adj}$ < 0.05, two-sided Wilcoxon rank-sum test, Supplementary Data S10) that were enriched for Club region markers (*Met A1* $p_{adj}$ = 3.2×10$^{-7}$ and *Met B5* $p_{adj}$ = 1.3×10$^{-13}$, one-sided Fisher's exact test) and *PMN-MDSC activity* markers (*Met A1* $p_{adj}$ = 3.2×10$^{-2}$, *Met B5* $p_{adj}$ = 2.3×10$^{-2}$, Fig. 5e). Genes overexpressed in the *Met B1* and *Met B2* clusters were similarly enriched for the Club region markers ($p_{adj}$ = 3.4×10$^{-7}$ and $p_{adj}$ = 1.5×10$^{-4}$, respectively). *Met B1* was enriched for all five of the tested signatures, including *EpiSen, club-like senescence,* and *high NLR-associated* genes. Canonical club cell marker *MMP7* was

**Fig. 4 | Club-like cell enriched regions are associated with increased myeloid-derived suppressor cell infiltration. a** GO:BP term enrichments for genes with upregulated expression in the Club (top) and Immune (bottom) regions. One-sided Fisher's exact test was used. **b** Correlation between Club region fraction and PMN-MDSC activity scores in pseudo-bulk for the discovery (left) and the validation (right) cohorts. Two-sided Spearman correlation coefficient and p-value are shown. Error bands in light grey span the 95% confidence interval calculated using a bootstrap. BPH: benign prostatic hyperplasia ($n = 4$), TRNA: treatment-naïve prostate cancer ($n = 17$); NEADT: neoadjuvant-treated prostate cancer ($n = 22$); CRPC: Castration-resistant prostate cancer ($n = 5$). Low cancer% $n = 11$, Mid cancer % $n = 11$, High cancer % $n = 10$. **c** PMN-MDSC activity score in treatment-naïve spots grouped according to their proximity to the Club region. Asterisks indicate two-sided independent samples t-test significance levels ($p < 0.05$) and effect size: *70th percentile <mean; **80th percentile <mean; ***90th percentile <mean. Club region

($n = 278$), proximal Tumor ($n = 687$), distant Tumor ($n = 8751$). **d** Violin plots of gene set activity scores across all regions in treatment-naïve ($n = 36,198$) and neoadjuvant-treated ($n = 59,259$) samples. Asterisks are the same as in (**c**). **e** A representative immunostained prostate tissue section (total $n = 16$) with selected regions of interest (ROIs). **f** Representative images of club-like positive ($n = 47$) and club-like negative ($n = 54$) ROIs corresponding to those annotated in (**e**). **g** A representative image of a club-like positive ROI ($n = 47$) with a three-colour stain showing CD66b$^+$CD11b$^+$CXCR2$^+$ PMN-MDSCs. Arrows point to example cases. Scale bars in white are 500 μm in each panel. **h** Scatterplot of log-transformed club-like cell and PMN-MDSC cell counts. Each dot represents a ROI ($n = 101$). Correlation coefficient, p-value, and error bands were calculated the same as in (**b**). **i** Violinplot of PMN-MDSC percentage of all detected cells in club-like negative ($n = 54$) and club-like positive ($n = 47$) ROIs. A Wilcoxon rank-sum test p-value is shown.

overexpressed in *Met B1* (log$_2$ foldchange = 1.06, $p_{adj} = 4.5 \times 10^{-24}$) and *Met B5* ($p_{adj} = 8.8 \times 10^{-21}$). *PIGR* was similarly overexpressed in *Met B5* ($p_{adj} = 1.2 \times 10^{-2}$). Other Club region markers overexpressed in *Met B5* included *CLDN1* ($p_{adj} = 7.8 \times 10^{-13}$), *SCNN1A* ($p_{adj} = 2.9 \times 10^{-6}$), and *TACSTD2* ($p_{adj} = 8.5 \times 10^{-61}$), all of which are exclusively expressed in epithelial cells. These results suggest the presence of club-like cells in metastatic prostate cancer tumors.

*Club-like senescence* and *PMN-MDSC activity* signature GSVA scores correlated across untreated primary and mCRPC tumors in the TCGA-PRAD and SU2C-PCF cohorts (Fig. 5g, h). The score of *club-like senescence* signature also correlated with two other MDSC signatures, and the *high NLR-associated* signature (Supplementary Fig. S10). *Club-like senescence* score correlated negatively with AR-signaling in both TCGA-PRAD ($r_s = -0.30$, $p = 1.2 \times 10^{-12}$, two-sided Spearman correlation test) and SU2C-PCF ($r_s = -0.28$, $p = 5.0 \times 10^{-6}$) cohorts.

Finally, we generated pseudo-bulk expression profiles of the discovery cohort ST samples to investigate how treatment affected the expression of senescence-related genes. Hierarchical clustering of the expression of *AR-signaling* and *club-like senescence* genes revealed nested sample groups within the cohort (Fig. 5f). Altogether 16 NEADT samples were characterized by decreased *AR-signaling* and increased *club-like senescence* activity. Differential expression analysis between the TRNA and NEADT groups revealed increased expression for genes in the *club-like senescence* and the *high NLR-associated* signatures (log$_2$ fold-change ≥ 1, $p_{adj} < 0.05$, two-sided Wald test, Fig. 5i, Supplementary Data S11). None of the genes in these signatures were downregulated post-treatment ($p_{adj} \geq 0.05$, two-sided Wald test).

## Discussion

We used spatial transcriptomics to explore the microenvironment of benign, untreated, neoadjuvant-treated, and castration-resistant prostate tumors. We used a single-cell expression reference to annotate tissue regions across 146,780 spatially defined data points, which not only recapitulated the histology annotated by pathologists but also allowed for joint analysis across a large set of ST data. Our analysis pipeline is inspired by NMF-based approaches that have captured recurrent gene expression programs across different tumor types[39,40,59] emphasizing gene set variation within individual samples before cross-sample integration. The advantage of our analysis is that we avoid batch effect issues related to commonly used clustering workflows[60,61], resulting in an unbiased cell-state reference.

Here we report that club-like cells have upregulated expression of genes encoding for molecules secreted in SASP[51], including TME-altering proteases *cathepsin β* (*CTSB*), *stromelysin-2* (*MMP10*), and *urokinase* (*PLAU*), *insulin-like growth factor-binding protein 3* (*IGFBP3*), and *intercellular adhesion molecule 1* (*ICAM1*). Furthermore, club-like cells display elevated expression of similarly SASP-associated myeloid cell chemotaxis-inducing chemokines[51,53] *CXCL1, CXCL2, CXCL3, CXCL8*, and *CCL20*, as well as the transmembrane chemokine *CXCL16*. In a

prostate-specific context, the expression of *CXCL1, CXCL2*, and *CXCL8* is independently predictive of poor overall survival in metastatic CRPC patients[45]. High *CCL20* expression in the prostate is associated with more aggressive disease[62] and its blockade slows tumor progression in mice[12]. In vitro studies have shown that *CXCL16* expression induces tumor cell and mesenchymal stem cell migration, promoting cancer metastasis to the bone[63,64]. We observed upregulated *CXCL12* expression in the Fibroblast region, and high *CXCR4* and *CXCR6* expression in the Immune region, all of which have been indicated as key molecules in possible metastasis mechanisms acting through *CXCL16*[64].

Increased cytokine signaling activity in the Club region is in line with reports that club-like cells are enriched in proliferative inflammatory atrophy (PIA)[24,65,66]. PIA has been regarded as a potential precursor for prostate cancer[65], with DNA damage resulting from cytokine-induced oxidative stress reported as one possible mechanism[14,67]. We found increased expression of PIA gene markers[66] *KRT5, KRT8*, and *MET*, and PIA-associated club-like cell gene markers[17,24] *CP, MMP7, LTF*, and *PIGR* in the Club region. Inflammation-related molecular signaling routes *IL6/JAK/STAT3, IFNγ response*, and *TNFα signaling* via *NFκB* also had increased activity in the Club region. Similar inflammatory signaling activity has been reported in club-like cells of histologically benign glands proximal to invasive cribriform carcinoma or intraductal carcinoma-enriched regions[9]. Inflammation-associated low-CD38 expressing luminal cells have previously been associated with biochemical recurrence[68], whereas high IFNγ signaling is associated with worse outcomes in high-risk localized disease[69]. These results indicate that the club-like cells are interchangeably linked with inflammation in the prostate TME.

Our findings corroborate a link between club-like cells and the luminal progenitor phenotype reported in mice[27]. Luminal progenitor cells are an inherently castration-resistant cell population in the mouse prostate[25,70] that have also been proposed as a possible cellular origin of prostate cancer[71]. These cells are mainly present in the proximal region of the prostatic duct, with scattered cells found in the distal lobes[20,23,72]. The previously described transcriptomic similarity of the mouse progenitor and human club-like cells is supported by our data[36]. Our analysis also shows that the Club region is unperturbed by ADT, indicating that human club-like cells are similarly resistant to castration. High expression of transcription factors *SOX9, KLF5*, and *ELF3* in the Club region is also indicative of the stem-like characteristics of these cells[73].

Finally, we postulate a connection between club-like cells and clinically significant immunosuppression. Chronic inflammation in the prostate facilitates cancer progression and results in the accumulation of pathologically activated neutrophils or PMN-MDSCs[14,46,74]. Immunosuppressive PMN-MDSCs are a driver of CRPC that can reactivate androgen signaling through IL-23 secretion[43]. In general, tumor sites contain expanded PMN-MDSC populations compared to healthy tissue[74,75], and the inhibition of their chemotactic receptor *CXCR2* can

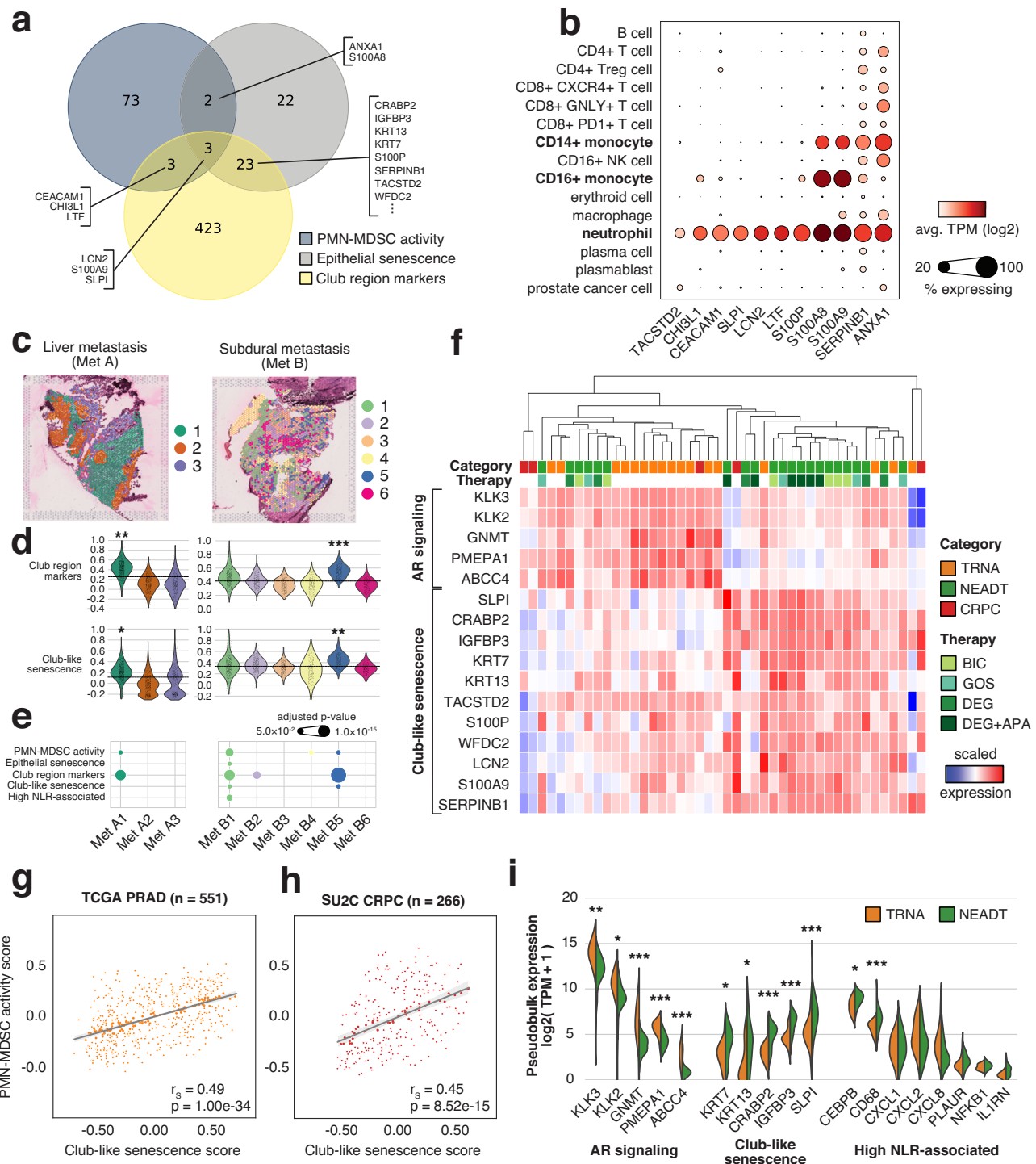

**Fig. 5 | Club-like senescence is associated with immunosuppressive PMN-MDSC activity in primary and metastatic tumors. a** Venn diagram showing overlaps between PMN-MDSC activity, epithelial senescence, and club-region upregulated gene sets. **b** Dot plot of normalized gene expression in metastatic castration-resistant prostate cancer samples. Data and clusters as reported in He et al. 2021[58]. **c** Expression-based clustering overlaid on two prostate cancer metastasis ST samples. **d** Cluster-specific gene set scores in MET A ($n = 2190$) and MET B ($n = 2346$) ST spots. The dashed line marks the overall score median. Asterisks indicate two-sided independent samples t-test p-value ($p < 0.05$) and effect size (*cluster 30th percentile > overall median; **cluster 20th percentile > overall median; *cluster 10th percentile > overall median) **e** Log-transformed overrepresentation $p_{adj}$ among each clusters overexpressed genes. One-sided Fisher's exact test was used. **f** Scaled

gene expression of pseudo-bulk spatial transcriptomics samples. **g, h** Score correlation for the Club-like senescence and PMN-MDSC activity signatures in TCGA PRAD ($n = 551$) and SU2C ($n = 266$) cohorts. Two-sided Spearman correlation coefficient and p-value are shown. Error bands in light grey span the 95% confidence interval calculated using a bootstrap. **i** Violin plot of normalized gene expression in pseudo-bulk spatial transcriptomics samples. Asterisks indicate differential gene expression test significance levels between treatment-naïve ($n = 17$) and neoadjuvant-treated ($n = 22$) prostate cancer samples (*$p_{adj}$ <0.05, **$p_{adj}$ < 0.01, ***$p_{adj}$ < 0.001, two-sided Wald test). PMN-MDSC: polymorphonuclear myeloid-derived suppressor cell; TRNA: treatment-naïve prostate cancer ($n = 17$), NEADT: neoadjuvant-treated prostate cancer ($n = 22$); CRPC castration-resistant prostate cancer ($n = 5$), BIC bicalutamide, GOS goserelin, DEG degarelix, APA apalutamide.

reverse castration resistance[45]. We show that club-like cells have elevated expression of canonical neutrophil chemokines and that their presence is associated with increased PMN-MDSC activity in primary tumors. Furthermore, we identify club-like cell populations in metastatic prostate cancer tumors, expanding their relevance outside the primary tumor setting.

Taken together, our analysis highlights club-like cells as a key TME constituent that is associated with PMN-MDSC infiltration. The combination of their ability to persist throughout ADT and to induce PMN-MDSC chemotaxis implicates them as a contributor to treatment resistance. Our findings warrant further research into how club-like cells contribute to prostate cancer initiation, progression, and resistance to therapy.

## Methods

### Sample collection

This study was carried out in compliance with all relevant ethical regulations and guidelines. The collection of patient samples was approved by the local ethical committees in Tampere University Hospital, Universitaire Ziekenhuizen KU Leuven, St. Olav's Hospital in Trondheim, and the Johns Hopkins Hopsital. For the discovery cohort, a total of 48 prostatectomy samples were collected at Tampere University Hospital (37) and UZ Leuven (11). Freshly frozen prostate tissue samples from 37 patients of different disease stages, including benign prostatic hyperplasia ($n = 4$), treatment-naïve ($n = 17$) and neoadjuvant-treated primary PCa ($n = 11$), locally recurrent castration-resistant prostate cancer (CRPC, $n = 5$) were obtained from Tampere University Hospital (Tampere, Finland) and utilized for the analysis. The neoadjuvant-treated samples were collected as part of clinical trial NCT00293696 evaluating the molecular effects of anti-androgen and chemical castration treatment[76]. The use of clinical material was approved by the Ethics Committee of the Tampere University Hospital and the National Authority for Medicolegal Affairs. For prospective sample collection, written informed consent was obtained from all the subjects. 11 freshly frozen neoadjuvant-treated primary PCa samples from radical prostatectomies samples of 11 patients were collected at the Universitaire Ziekenhuizen KU Leuven as part of clinical trial NCT03080116[77]. The mean age of discovery cohort cancer patients at surgery was 63 years (range: 33–79), the mean ISUP grade group was 2.7 (range: 1–5), and the mean pre-operation blood PSA level 12.7 ng/ml (range: 3.2–111) (Supplementary Data S1).

Of the metastatic prostate cancer samples used in this study, three were acquired as part of the Johns Hopkins Medicine Institutional Review Board-approved (NA_00003925) Project to ELIminate lethal CANcer (PELICAN) from patients who provided written informed consent. One sample was acquired under Tampere University Hospital Ethics Committee approval R19074 from a patient who had provided written informed consent.

For the validation cohort, 8 freshly frozen prostate tissue specimens were collected from patients with untreated primary PCa who had given informed written consent before undergoing radical prostatectomy at St. Olav's Hospital in Trondheim between 2008 and 2016. This research received approval from the regional ethical committee of Central Norway (identifier 2017/576) and adhered to both national and EU ethical regulations. The mean age at surgery was 59 years (range: 53–73), the mean ISUP grade was 3.4 (range: 2–5) and the mean pre-operation blood PSA level was 20.1 ng/ml (range: 9.6–45.9). Three patients remained relapse-free for >10 years following surgery, while five patients experienced relapse with confirmed metastasis within three years. For each patient, two samples with cancerous tissue (Cancer), one sample with noncancerous morphology close to the cancerous region (Field effect), and one sample with noncancerous morphology distant from the cancerous area (Normal) were sectioned. In total, 32 samples (no relapse $n = 12$, relapse $n = 20$) were collected.

### Spatial transcriptomics assay

For the discovery cohort and the metastatic tumor samples, tissue sections were profiled for spatial transcriptomics using the Visium Spatial Gene Expression Reagent Kit protocol from 10x Genomics (CG000239, Rev F, 10x Genomics). The tissues were cryosectioned with Cryostat SLEE MEV+ at 10 μm thickness to the Visium spatial gene expression slide, fixed in ice-cold 100% methanol for 30 min at −20 °C, and hematoxylin and eosin (H&E) staining was done manually based on the Methanol Fixation, H&E Staining & Imaging for Visium Spatial Protocols (CG000160, Rev D, 10x Genomics). The capture areas were imaged individually using Hamamatsu NanoZoomer S60 digital slide scanner. Sequencing library preparation was performed according to the Visium Spatial Gene Expression user guide (CG000239 Rev F, 10x Genomics) using a 20-minute tissue permeabilization time. Sequencing was done on the Illumina NovaSeq 6000 sequencer at Novogene Company Limited, (Cambridge, UK) or CeGaT GmbH (Tübingen, Germany) sequencing facilities, aiming at a minimum of 50,000 read pairs per tissue-covered spot (55 μm; 1–10 cells) as recommended by the manufacturer.

For the validation cohort samples, 10 μm-thick tissue sections were cut at −20 °C using a Cryostar NX70 (Thermo Fisher), placed onto the Visium slides, and processed following the manufacturer's protocol. Before extracting RNA, tissue sections were treated with 100% methanol, stained with H&E, and scanned digitally at 20x magnification. RNA was isolated by treating the tissue sections with a permeabilization agent for 12 minutes. The most effective extraction time had been established earlier using the Visium Spatial Tissue Optimization Slide & Reagent kit (10X Genomics). Following this, a second strand mix was introduced to generate a complementary strand, after which cDNA was amplified using real-time qPCR. The resulting amplified cDNA library was quantified using the QuantStudio™ 5 Real-Time PCR System (Thermo Fisher) via qPCR, and the cDNA libraries were preserved at −20 °C until needed for subsequent experiments.

### Histopathology evaluation

For the discovery cohort samples that were generated at Tampere, a board-certified uropathologist (T.M.) assigned individual spots into one of 9 categories (Gleason 5, Gleason 4,

Gleason 4 cribriform, Gleason 3, Atrophy, Prostatic Intraepithelial Neoplasia (PIN), Inflammation, Benign, and Stroma) using Loupe Browser version 6.0 (10x Genomics). Quality control of selected samples and qualitative assessment of histological concordance with SCM regions was performed by an additional pathologist (C.T.A.P.).

For the validation cohort, two experienced uropathologists (T.V. and Ø.S.) independently evaluated the H&E-stained sections from all 32 spatial transcriptomics samples in QuPath (version 0.2.3). They annotated cancer areas according to the International Society for Urological Pathology (ISUP) Grade Group system and also identified aggregates of lymphocytes. For the downstream data analysis, a consensus pathology annotation was reached in agreement with both pathologists. Gleason scores were transformed into ISUP grade groups (ISUP1–5). Cancer areas with uncertain grading were annotated as ISUPX (indecisive between ISUP3 and ISUP5) and ISUPY (indecisive between ISUP1 and ISUP4). Other section annotations included non-cancerous glands, stroma, and stroma with higher levels of lymphocytes (referred to as 'lymphocyte-enriched stroma'). A detailed description of how histology annotations in QuPath were converted into spot-wise histology classes is given elsewhere[78].

### Spatial transcriptomics data preprocessing

Libraries were sequenced with NovaSeq 6000 PE150. Images of each capture area were transformed from ndpi to tiff-format at the 20x zoom level using *ndpi2tiff v1.8*[79]. Tissue alignment masks were created using Loupe Browser v4.2.0. Read alignment to GRCh38 and transcript counting were performed using spaceranger v1.1.0, where the tiff

image and the tissue alignment masks were also used as input. The results were read into Python 3.7 using *scanpy.read_visium* (scanpy v1.9.1). For each sample individually, genes present in less than 5 spots and spots with less than 500 UMIs were discarded using *scanpy.pp.filter_genes* and *scanpy.pp.filter_cells*, respectively. Transcript counts were normalized using the single-cell integration benchmark (scib, v1.1.1) implementation of the scran method by running *scib.preprocessing.normalize*[80,81].

## Single-cell reference mapping

The scRNA-seq cell-state reference created from publicly available data[6–9,12,17,32] (Supplementary Methods) was transformed into a mappable reference using *cell2location.models.RegressionModel* (cell2location v0.1.3) with the dataset of origin used as *batch_key*, cell-state annotation as *labels_key*, and original sample used as *categorical_covariate_key*. The regression was performed on unnormalized gene counts. The model was trained for 250 epochs on an NVIDIA Tesla V100 16GB graphics processing unit (GPU). *RegressionModel.export_posterior* with parameters *num_samples: 1000, batch_size: 2500* on the same GPU was used to export the posterior distribution. From these distributions, the mean expression of each gene in each cell state (*means_per_cluster_mu_fg*) was used for mapping. A single cell2location model was created using *cell2location.models.Cell2location* by concatenating the untransformed (raw) ST data gene counts and setting the parameter *batch_key* as the sample identifier. The model was trained on the same NVIDIA Tesla V100 GPU with the following parameters: *N_cells_per_location = 21, detection_alpha = 20, max_epochs = 30000, batch_size = 34000, train_size = 1*. The inferred cell abundances for each spatial location across all samples were exported using *export_posterior* with *num_samples = 1000*.

## Defining single-cell mapping-based regions

The 5th quantile of each cell state's abundance distribution (*q05_cell_abundance_w_sf*) was used as the inferred cell count as per the developer's instructions (Supplementary Fig. S11). These cell abundances were treated as an alternative to gene counts, where the feature space was defined as 26 cell states. We determined that not all cell states were relevant in terms of tissue organization, and set out to define the smallest number of regions that were 1) ubiquitous, i.e. present in all sample classes, and 2) biologically meaningful.

Non-negative matrix factorization (NMF) as implemented in *scikit-learn* v1.1.3 was used to reduce the feature space. Iterative NMF with an increasing number of components (5 to 12) was run, stopping at the highest number of components where each factor had a unique cell state as the highest-contributing feature. The iteration with the highest number of components where no single cell state was the highest contributor in multiple components was chosen as the optimal iteration (*n_components = 8*, Supplementary Fig. S12). Each ST data point was then categorized according to its highest contributing component, resulting in eight regions across the whole ST dataset (Supplementary Fig. S13).

## Region-specific gene markers

For each sample individually, scanpy's *rank_genes_groups* function with *method = 'wilcoxon'* was used to calculate differentially expressed genes between the SCM regions, where the expression of each gene in each region was compared to the baseline expression of that gene in other regions of the sample. Regions with fewer than 10 members and genes of ribosomal or mitochondrial origin were excluded from the analysis. For each region, genes with log fold change $\geq 1$ and $p_{adj} < 0.05$ were considered differentially expressed. A one-sided Fisher's exact test was then used to test whether a gene was enriched as a marker for a specific region. The test was carried out by comparing the number of instances where the gene was differentially expressed in a region of interest against the number of instances where the gene was differentially expressed in any other region. Genes with $p_{adj} < 0.05$ were considered as region-specific markers. A single gene could be considered a region-specific marker for multiple regions.

## Neighborhood enrichment analysis

Sample-specific region neighborhood enrichments were calculated using *squidpy* v1.2.3. First, a neighborhood graph was built for each sample individually using *squidpy.gr.spatial_neighbors* with parameters *coord_type = "grid", n_neigh = 6*, and *n_rings = 3*. *squidpy.gr.nhood_enrichment* was then used with default parameters to calculate the enrichment scores. This score is based on a permutation test where the association between regions is estimated by comparing the true region layout to a randomly sampled configuration. Enrichment scores of missing regions were set to 0 in each sample. The mean of all scores was calculated across all samples to get dataset-wide enrichment scores.

## Spot-level gene set activity scoring and gene set enrichment analysis

Scanpy implementation of Seurat's scoring method in *scanpy.tl.score_genes* was used to score 154 gene sets on the normalized data[82,83]. Parameter values *ctrl_size: 50, n_bins: 25* were used. No genes were excluded from the control gene pool. Only genes that passed the quality control were included in calculating the score for each sample. A one-sided Fisher's exact test was used to perform gene set enrichment analysis of custom gene sets on the region-specific gene markers. Gene sets with $p_{adj} < 0.05$ were considered enriched. g:Profiler was used to perform enrichment analysis of region-specific gene markers on the Gene Ontology: Biological Process database[84,85].

## Region fraction and gene signature score correlation analysis

Each sample's un-normalized transcript counts were concatenated while leaving out spots that were annotated as the region of interest. Genes expressed in fewer than 10 samples were discarded, after which the data was normalized using *scanpy.pp.normalize_total* and log-transformed with *scanpy.pp.log1p*. Gene set activity scores were calculated identically to the spot-level scores. A Spearman correlation coefficient and a corresponding p-value were calculated for the Club region fraction and gene set scores in each sample.

## Multiplex immunohistochemistry staining

Formalin-fixed paraffin-embedded prostate tumor tissue blocks from 16 patients were obtained from Tampere University Hospital (Tampere, Finland). The use of clinical material was approved by the Ethics Committee of the Tampere University Hospital and the National Authority for Medicolegal Affairs. For prospective sample collection, written informed consent was obtained from all the subjects. The tumor samples were fixed in 4% phosphate-buffered formaldehyde and processed into paraffin blocks. 5 μm thick sections were stained with in-house multiplex-IHC protocol based on Multiple Iterative Labeling by Antibody Neodeposition (MILAN)[86]. Tissue sections were treated with heat-induced epitope retrieval (HIER) using Tris-HCl buffer (pH 9.0) prior to antibody labeling. Antigens were stained with anti-PIGR 1:500 (Sigma-Aldrich, HPA012012), anti-LTF 1:500 (Sigma-Aldrich, HPA059976), anti-Pan Keratin (AE1/AE3/PCK26) (Roche, 760-2135), anti-CP 1:150 (HPA001834, Sigma-Aldrich), anti-CD66b 1:50 (Novus Biologicals, NB100-77808), anti-CXCR2 1:2000 (Abcam, ab245982), anti-CD45 1:100 (Cell Signaling Technology, #13917) and anti-CD11b 1:100 (Cell Signaling Technology, #49420). Detection was done with fluorescently labeled secondary antibodies Goat anti-Rabbit IgG (H + L) Highly Cross-Adsorbed Secondary Antibody, Alexa Fluor Plus 647 (Invitrogen, A32733), Goat anti-Mouse IgG (H + L) Cross-Adsorbed Secondary Antibody, Alexa Fluor 750 (Invitrogen, A21037)). Nuclei were stained with DAPI (4',6-Diamidino-2-Phenylindole, Dihydrochloride) (Invitrogen, D1306) and slides were mounted using

Fluorescence Mounting Medium (Agilent, S3023). The staining result was scanned using NanoZoomer S60 (Hamamatsu) whole-slide scanner. The fluorescent multiplex-IHC staining was followed by an HE-staining on the same tissue section. HE-staining was done using Leica ST5010 Autostainer XL and the slides were mounted using DPX mountant (Sigma-Aldrich, 44581).

## Multiplex immunohistochemistry image processing

The DAPI staining images were overlaid (registered) utilizing in-house scripts and Python package VALIS (version 1.0.0rc13)[87]. DAPI staining image of LTF-panCK was used as a fixed reference for each staining round. The transformation matrices obtained from these were applied to each matching staining image to obtain the same orientation. Regions of interest (ROI) were obtained by selecting representative areas from the tissue and re-selecting those from subsampled (long side 2048px) images using Qupath (version 0.4.4) TMA dearrayer (width 120px)[88]. These obtained coordinates were transformed to original images to crop full-size ROIs.

Images were processed into hyperstacks using Fiji-ImageJ version v1.53[89]. Image analysis of registered images was performed in Qupath version 0.4.4. Individual cells were segmented from images using Stardist (model dsb2018_heavy_augment)[90]. Cell classifier of seven cell categories (Club-like cells (panCK + , LTF+ and/or PIGR + ), CP+ Club-like cells (panCK + , CP + ), panCK+ cells (panCK + ), Granulocytes (CD45 + , CD11b + , CD66b + ), MDSCs (CD45 + , CD11b + , CD66b + , CXCR2 + ), Other cells (negative to all except DAPI) and Other immune cells (CD45 + ) (Rtrees) was created based on 26 training images selected from across samples in the staining cohort and applied to a total of 101 ROIs. Classifier results were exported as cell annotation counts and further analyzed in Python.

## Region interface annotation and ligand-receptor analysis

Spatial neighborhood graphs were constructed for each sample as in the neighborhood enrichment analysis. With the Club region selected as the region of interest, the region identities of neighboring spots were surveyed to find another spot of the same annotation. This was done to exclude sporadic individual spots without neighbors of the same region. All non-club spots within 3 rings (300 μm) of two or more adjacent Club region spots were then annotated as 'proximal', and all spots further than 3 rings away as 'distant'. The previously calculated gene set scores in these spot categories were then compared across all samples. 6 samples with fewer than 10 Club region-annotated spots were excluded from the analysis. For region-specific interactions (Fig. 4d), the clause that a spot was to have a neighboring spot of the same region identity was included for the interacting region. Additionally, each spot was required to have at least 2 spots from the interacting cluster within three spots distance to be considered as interacting. At least 10 spots from both regions were required in a sample for it to be used in the ligand-receptor interaction analysis. Interfaces between Club and Endothelium were not considered for downstream analysis due to low prevalence (6).

*squidpy*'s implementation of the *CellphoneDB*[91] method was used to calculate ligand-receptor interactions at the sample level. *squidpy.gr.ligrec* was run between interface-labeled spots using parameter settings *complex_policy='all'*, *threshold = 0.01*, and *n_perms = 1000*. Ligand-receptor pairs with $p_{adj} < 0.05$ were considered active and were used in the overrepresentation analysis. Interactions with fewer than 3 literature references in the *OmniPath*[92] database were discarded.

## Analysis of metastatic CRPC scRNA-seq data

Single-cell gene expression data of mCRPC samples were downloaded from a public repository (https://singlecell.broadinstitute.org/single_cell/study/SCP1244/transcriptional-mediators-of-treatment-resistance-in-lethal-prostate-cancer). TPM normalized counts were matched with the cell identity annotation as defined in the original analysis. Gene expression was visualized using *scanpy.pl.dotplot*.

## Spatial transcriptomics data analysis of metastatic prostate cancer tumors

Three of the four metastatic tissue samples studied were collected as part of the PELICAN integrated clinical-molecular autopsy study of lethal prostate cancer (PELICAN)[93]. Subjects A14 (Met A), A3 (Met C), and A16 (Met D) included in this study provided written informed consent to participate in the Johns Hopkins Medicine IRB-approved study between 1995 and 2005. The mean age of the study subjects at the time of diagnosis of prostate cancer was 61 years.

Sample GP12 (Met B) was collected as a part of the Geoprostate study, where patients newly diagnosed with PrCa electing radical prostatectomy (RP) with 20% or greater preoperative risk of pelvic lymph node metastasis were eligible to participate under Tampere University Hospital Ethics Committee approval R19074 and provided written informed consent to participate in the study[94].

ST libraries were generated identically to the discovery cohort primary tumor samples generated in Tampere. An identical data pre-processing procedure was followed. Expression clusters were generated using the standard clustering workflow outlined in the *scanpy* (v1.9.1) manual. The data was scaled using *scanpy.pp.scale()*, followed by *scanpy.tl.pca()*, *scanpy.pp.neighbors()*, and *scanpy.tl.leiden()* with a parameter setting *resolution = 0.5*. All the other parameters were used in their standard setting. Differential gene expression analysis was performed by running *scanpy.tl.rank_genes_groups()* on the *leiden* clusters and with *method=wilcoxon*. Overexpressed genes in each cluster were defined as having $p_{adj} < 0.05$ (two-sided Wilcoxon rank-sum test) and $log_2$ *fold-change* ≥ 1. These genes were used as input in the enrichment analysis. Gene set activity scores were calculated for each data point with *sc.tl.score_genes()* on the normalized but unscaled expression counts.

## Analysis of bulk transcriptomics data

TCGA-PRAD expression data was downloaded from a public repository (https://tcga.xenahubs.net). FPKM normalized polyA mRNA expression data of the SU2C-PCF mCRPC cohort was downloaded from a public repository (https://www.cbioportal.org/study/summary?id=prad_su2c_2019)[95]. *gseapy.gsva* (v1.1.0) was used to calculate GSVA scores for each sample[96]. Possible gene overlap was removed when calculating scores for the correlation analysis of two signatures.

## Spatial transcriptomics pseudo-bulk GSVA and differential expression analysis

Each ST sample's un-normalized transcript counts were concatenated to form pseudo-bulk expression matrices. Gene counts were TPM normalized and *gseapy.gsva* (v1.1.0) was used to calculate GSVA scores for each sample. Possible gene overlap was removed when calculating scores for the correlation analysis of two signatures. Differential expression analysis between TRNA and NEADT samples was performed using *pydeseq2* (v0.4.4)[97] with TRNA used as the reference level.

## Statistics and reproducibility

*scipy.stats* v1.9.3 and *statsmodels.stats.multitest* v0.13.5 in python 3.8 was used to perform statistical testing and multiple testing corrections, respectively. The Benjamini-Hochberg method was used to adjust p-values for multiple testing. No statistical method was used to predetermine sample size. No data were excluded from the analysis, except for the spatial transcriptomics data points that did not meet the criteria for minimum number of UMIs or genes. The experiments were not randomized. The investigators were not blinded to allocation during experiments and outcome assessment.

**Reporting summary**

Further information on research design is available in the Nature Portfolio Reporting Summary linked to this article.

## Data availability

The processed spatial transcriptomics data presented in this study, excluding the validation cohort, have been deposited in the Gene Expression Omnibus (GEO) archive under accession identifier GSE278936. The raw sequencing data are available from the authors, but restrictions apply to the availability of these data. Data can be shared with qualified researchers in accordance with the conditions of ethical approvals and informed consent to use these data in research of prostatic diseases. All handling of these data must be in compliance with GDPR and other relevant data protection regulations upon completion of material transfer agreement with respective data controllers' information. Data access requests will be processed at the earliest convenience. Data access will be granted for one year. Validation cohort spatial transcriptomics data is available in the European Genome-Phenome Archive (EGA) under accession identifier [EGAD50000000603]. The data is available under restricted access to ensure compliance with ethical and legal standards, including GDPR and approval from the Regional Committee for Medical Research in Norway. Access will be granted to researchers who meet these requirements, and they must sign a Data Access Agreement (DAA). To obtain access, contact the Data Access Committee (DAC) at NTNU, who will facilitate the process and provide access through the FEGA Norway node or HUNT Cloud once the DAA is completed. The DAA will be processed at the earliest convenience. The single cell RNA-sequencing datasets used in this study[6–9,12,17,32] are available on GEO under accession numbers GSE137829, GSE141445, GSE176031, GSE185344, and GSE181294, the Sequence Read Archive (SRA) under accession number PRJNA699369 and on https://singlecell.broadinstitute.org/single_cell/study/SCP1244/transcriptional-mediators-of-treatment-resistance-in-lethal-prostate-cancer[58]. The bulk RNA-sequencing data used in this study is available for download on https://tcga.xenahubs.net (TCGA)[3] and on https://www.cbioportal.org/study/summary?id=prad_su2c_2019[93]. The remaining data are available within the Article, Supplementary Information or Source Data file. Source data are provided with this paper.

## Code availability

Code used for data analysis in this manuscript is available on https://github.com/akiviaho/ST-prostate.

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

## Acknowledgements

The authors acknowledge the Biocenter Finland (BF) and Tampere Genomics Facility for their service. The authors wish to acknowledge CSC–IT Center for Science, Finland, for computational resources. The results published here are in part based upon data generated by The Cancer Genome Atlas project (dbGaP Study Accession: phs000178.v9.p8) established by the NCI and NHGRI. Information about TCGA and the investigators and institutions who constitute the TCGA research network can be found at http://cancergenome.nih.gov. Tissue samples of the validation were collected and stored by Biobank1, St. Olav's Hospital. Tissue sectioning, staining, and scanning were performed by or in collaboration with the Histology lab at the Cellular & Molecular Imaging Core Facility at NTNU, while RNA isolation and sequencing were carried out at the Genomics Core Facility at NTNU. We thank Sari Toivola, Päivi Martikainen, Hanna Selin, and Maria Annala for their assistance in laboratory procedures. The authors wish to thank all participating patients and their families. A.K. has received funding from the Tampere University Doctoral School, the Finnish Cultural Foundation, and the Cancer Foundation Finland. C.T.A.P. has received funding from the Jean Shanks Foundation and the Pathological Society Clinical PhD Fellowship. S.T. has been funded by the Tampere University Doctoral School and the Relander Foundation. E.A. has received funding from the Finnish Cultural Foundation. T.M. has received funding from the Finnish Cancer Institute, Helsinki University Hospital State Funding (VTR), the Sigrid Juselius Foundation, and the Cancer Foundation Finland. M.B.T., M.K.A., S.Krossa., M.B.R., M.W., E.M., G.F.G., and A.S. have collectively benefited from funding provided by the European Research Council (ERC) under the Horizon 2020 program, the Liaison Committee between the Central Norway Regional Health Authority (RHA) and NTNU, the Norwegian Cancer Society, and the Terje Eugen Johnsen funds. K.J.R. has been funded by the Cancer Foundation Finland, the Sigrid Juselius Foundation, the Emil Aaltonen Foundation, the Competitive State Research Financing of the Expert Responsibility area of Tampere University Hospital, the Väre Research Foundation, and the Aamu Pediatric Cancer Foundation. A.U. has received support from the Academy of Finland (project no. 349314), the Cancer Foundation Finland, the Norwegian Cancer Society (project no. 198016-2018 & project no. 273672-2023), and the Tampere Institute for Advanced Study. M.N. has been funded by the Academy of Finland Center of Excellence program (project no. 312043), the Cancer Foundation Finland, the Sigrid Juselius Foundation, and the Competitive State Research Financing of the Expert Responsibility area of Tampere University Hospital.

## Author contributions

K.J.R., A.U., and M.N. supervised this work. A.K., K.J.R. A.U., and M.N. conceived and designed the analysis. A.K. designed the figure panels and wrote the original manuscript draft. S.K.E., H.M.L.K., M.K.A., M.H., A.M.T., E.M., K.V., and S. Kint performed experimental studies. A.K., A.M.T., I.S., S.T., O.H., M.I., M.W., A.S., S. Krossa, T.H., E.A., J.K., and M.B.R. performed computational analyses and contributed data analysis tools. C.T.A.P., T.T., T. Viset, Ø.S., and T.M. performed histopathology annotation. X.S., A.G., W.D., T.L.J.T., T.M., G.S.B., S.J., J.V.S., T. Visakorpi, G.A., F.C., T. Voet., K.J.R., and M.B.T. contributed data and provided resources for the study. A.K., S.K.E., H.M.L.K., M.K.A., A.G., C.T.A.P., M.M., I.H., G.F.G., M.B.R., T.M., A.E., L.L., G.S.B., I.G.M., S.J., T.M., G.A., F.C., T. Visakorpi, K.J.R., M.B.T., A.U., and M.N. reviewed and edited the manuscript. All authors reviewed and approved the final version of the manuscript.

## Competing interests

C.T.A.P.'s employer may gain commercially from licensing data to Artera AI. G.A. received personal fees, grants, and travel support from Janssen and Astellas Pharma; personal fees or travel support from Pfizer, Novartis/AAA, Bayer Healthcare Pharmaceuticals, AstraZeneca, and Sanofi-Aventis; in addition, G.A.'s former employer, The Institute of Cancer Research, receives royalty income from abiraterone and G.A. receives a share of this income through the Institute's Rewards to Discoverers Scheme. G.A. has received research funding (institutional) from Janssen, Astellas Pharma, and Novartis. All other authors declare no potential conflicts of interest.

## Additional information

Antti Kiviaho [1,2], Sini K. Eerola[1,2], Heini M. L. Kallio [1,2], Maria K. Andersen [3,4], Miina Hoikka [1,2], Aliisa M. Tiihonen [1,2], Iida Salonen [1,2], Xander Spotbeen[5,6], Alexander Giesen [7,8], Charles T. A. Parker[9], Sinja Taavitsainen [1,2], Olli Hantula[1,2], Mikael Marttinen[1,2,10,11], Ismaïl Hermelo[1,2], Mazlina Ismail [9], Elise Midtbust[3,4], Maximilian Wess [3,4], Wout Devlies [7,12], Abhibhav Sharma [13], Sebastian Krossa [3,14], Tomi Häkkinen[1,2], Ebrahim Afyounian [1,2], Katy Vandereyken[6,15], Sam Kint [6,15], Juha Kesseli [1,2], Teemu Tolonen [2,16], Teuvo L. J. Tammela[1,17], Trond Viset[18], Øystein Størkersen[18], Guro F. Giskeødegård [4,13], Morten B. Rye[4,19], Teemu Murtola[1,2], Andrew Erickson[20,21,22], Leena Latonen [23], G. Steven Bova [1,2], Ian G. Mills[22,24], Steven Joniau [7,8], Johannes V. Swinnen [5,6], Thierry Voet [6,15], Tuomas Mirtti[20,21,25], Gerhardt Attard [9,26], Frank Claessens [12], Tapio Visakorpi[1,2,27], Kirsi J. Rautajoki [1,2] May-Britt Tessem [3,4], Alfonso Urbanucci [1,2,28,29] ✉ & Matti Nykter [1,2,29] ✉

[1]Faculty of Medicine and Health Technology, Tampere University, Tampere, Finland. [2]Prostate Cancer Research Center, Tampere University and TAYS Cancer Center, Tampere, Finland. [3]Department of Circulation and Medical Imaging, Norwegian University of Science and Technology, Trondheim, Norway. [4]Clinic of Surgery, St. Olavs Hospital, Trondheim University Hospital, Trondheim, Norway. [5]Laboratory of Lipid Metabolism and Cancer, KU Leuven and Leuven Cancer Institute (LKI), Leuven, Belgium. [6]KU Leuven Institute for Single Cell Omics (LISCO), KU Leuven, Leuven, Belgium. [7]Department of Urology, University Hospitals Leuven, Leuven, Belgium. [8]Department of Development and Regeneration, KU Leuven, Leuven, Belgium. [9]University College London Cancer Institute, London, UK. [10]European Molecular Biology Laboratory, Structural and Computational Biology Unit, Heidelberg, Germany. [11]European Molecular Biology Laboratory, Genome Biology Unit, Heidelberg, Germany. [12]Molecular Endocrinology Laboratory, Cellular and Molecular Medicine, KU Leuven, Leuven, Belgium. [13]Department of Public Health and Nursing, Norwegian University of Science and Technology (NTNU), Trondheim, Norway. [14]Central staff, St. Olavs Hospital HF, 7006 Trondheim, Norway. [15]Laboratory of Reproductive Genomics, Department of Human Genetics, KU Leuven, Leuven, Belgium. [16]Department of Pathology, Fimlab Laboratories, Ltd, Tampere University Hospital, Tampere, Finland. [17]Department of Urology, Tampere University Hospital, Tampere, Finland. [18]Department of Pathology, St. Olav's Hospital, Trondheim University Hospital, Trondheim, Norway. [19]Department of Clinical and Molecular Medicine, Norwegian University of Science and Technology (NTNU), Trondheim, Norway. [20]Research Program in Systems Oncology, Faculty of Medicine, University of Helsinki, Helsinki, Finland. [21]ICAN-Digital Precision Cancer Medicine Flagship, Helsinki, Finland. [22]Nuffield Department of Surgical Sciences, University of Oxford, Oxford, UK. [23]Institute of Biomedicine, University of Eastern Finland, Kuopio, Finland. [24]Patrick G Johnston Centre for Cancer Research, Queen's University of Belfast, Belfast, UK. [25]Department of Pathology, University of Helsinki & Helsinki University Hospital, Helsinki, Finland. [26]University College London Hospitals, London, UK. [27]Fimlab Laboratories, Ltd, Tampere University Hospital, Tampere, Finland. [28]Department of Tumor Biology, Institute for Cancer Research, Oslo University Hospital, Oslo, Norway. [29]These authors jointly supervised the work: Alfonso Urbanucci, Matti Nykter. ✉e-mail: alfonso.urbanucci@tuni.fi; matti.nykter@tuni.fi

