## [Transparent Peer Review file · Nature Communications]

Single cell and spatial transcriptomics highlight the interaction of club-like cells with immunosuppressive myeloid cells in prostate cancer

Corresponding Author: Professor Matti Nykter

Version 0:

Reviewer comments:

Reviewer #1

(Remarks to the Author)

Kiviaho A et al did an extensive spatial profiling analyses of prostate cancer cells and their associated microenvironment using 80 sections from 56 patients containing treatment naïve BPH and prostate cancer tissues, neoadjuvant treated prostate cancer and castration resistant prostate cancer specimens. The authors show that club-like cells display a senescence-associated secretory phenotype and there is an increased MDSC cells proximal to club-like cell regions in the tumor specimens. Some potential ligand-receptor interactions between club-like cells and MDSC are also revealed through the study. The study is well designed with proper controls. The results are described clearly. The spatial transcriptomics analysis is comprehensive and provides novel information regarding the cellular compositions and cell-cell interactions in human prostate specimens. The novel information obtained from this study will benefit the prostate cancer research community and help advance our understanding of the disease. Below are my major and minor comments.

Major concerns

- (1) The conclusion is exclusively based on the spatial transcriptomic analysis. It would be nice if the authors could corroborate at least some of the major conclusions using a different technology, such as multiplex RNA-in-situ or IHC analysis. For example, the higher expression of certain chemokines at club-like cell regions. I understand that it is difficult to identify PMN-MDSC using single markers, but at least using some relevant markers reported in this manuscript such as S100A9 etc. would make the conclusion more convincing.
- (2) It would be interesting if the authors could check the expression profiles of club-like regions in different prostate cancer specimens and check if they can obtain additional information regarding how the expression profiles of the club-like cells are different in benign tissues versus tumor adjacent tissues and in treatment naïve tumors versus neoadjuvant-treated or castration resistant prostate cancer specimens.

Minor concerns

- (1) In the abstract the authors stated that the data were from 110 patients. In the first paragraph of the result section, the authors mentioned 56 patients. But the Method section only mentioned 45 patients.
- (2) The other marker for the club cell is PSCA. How the expression of PSCA is in the datasets?
- (3) Line 434, the authors should consider citing PMID: 26418304 when describing the presence of club cells in proximal region of mouse prostatic ducts.

(Remarks on code availability)

Reviewer #2

(Remarks to the Author)

Overview:

The manuscript presents a comprehensive cohort study that includes 10x Visium profiles of 80 human prostate tissues from 56 individual patients. To enhance the spatial data references, the authors integrated scRNA-seq data from seven previous

studies, which included both normal and tumor prostate tissue samples. Using Non-negative Matrix Factorization (NMF), eight single-cell mapping-based (SCM) regions were defined. All biological insights were derived at the level of these SCM regions. Overall, the manuscript is well-written and easy to follow. The study's large sample size ensures the validity of the biological results. The bioinformatics analyses were conducted properly, with detailed information on the software tools used, including their versions and functions.

Major comments:

1. The most significant results of this manuscript are based on the eight single-cell mapping-based (SCM) regions defined using comprehensive scRNA-seq references. To determine the optimal number of SCM regions, the authors performed iterative Non-negative Matrix Factorization (NMF) with different numbers of components, ranging from 5 to 12. They selected the smallest number of components where no single cell state was the highest contributor in multiple components, as illustrated in Figure S9. However, according to Figure S9, no single cell state was the highest contributor in multiple components when the number of components was 5, 6, or 7. It is unclear why 8 components were ultimately chosen. Given the frequent mention of these eight SCM regions throughout the manuscript, it is important to clarify the rationale behind this decision.

2. This is a follow-up to the previous point. Two of the eight SCM regions defined from the reference scRNA-seq datasets are muscle and fibroblast. However, as noted by the authors in Figure S12, "fibroblast and muscle categories were merged into one due to their similarity in terms of marker gene expression." If these two cell types are so similar, why do they appear as distinct SCM regions after NMF? Should these two SCM regions be considered as one? Some explanation and justification is needed here.

Minor comment:

1. The integration analysis of all the scRNA-seq reference datasets (98 samples from 64 patients) is crucial, as it forms the basis for defining the 8 SCM regions. However, all related results and figures are only found in the supplementary material. Including some of these in the main manuscript would make more sense.

(Remarks on code availability)

Reviewer #3

(Remarks to the Author)

In the present manuscript, Kiviahio et al. highlight the role of club-like epithelial cells in the tumor microenvironment of prostate cancer.

They propose that this subset of epithelial cells may be linked to the presence of tumor-infiltrating PMN-MDSCs. The authors utilized single-cell RNA-sequencing and spatial transcriptomics from prostate cancer samples of 110 patients.

Overall, the manuscript is technically solid and interesting. The strength of this work lies in the spatial transcriptomics data generated from valuable patient samples.

Previous studies, such as Henry et al. (Cell Reports 2018), have reported cells with the same signature as club cells in healthy and cancerous prostate tissue.

This manuscript adds to the current knowledge by showing that these cells express elevated levels of canonical neutrophil chemokines and are associated with increased PMN-MDSC activity in primary and metastatic tumors, an immune population already demonstrated to have pro-tumorigenic roles in CRPCs.

Although the manuscript is intriguing and the data are unique, there are some minor concerns to address:

1. The authors do not provide a clear cause-and-effect experiment to support the statement in the abstract that "club-like cells induce myeloid inflammation." It is suggested to revise this statement.
2. The manuscript indicates that the presence of various chemokines is a sign of SASP (line 407). However, the molecules found may be involved in too many mechanisms to be defined as SASP.
3. The conclusions are very definitive and should be softened. For example, the sentence on line 254, where the presence of transcripts for chemokines involved in neutrophil recruitment, is not proof of active neutrophil recruitment into the tumor.
4. Given that the strength of this work lies in the methodology used and the data generated, these should be made available to the scientific community. Currently, the reporting summary states: "The data that support the findings of this study are available from the authors but restrictions apply to the availability of these data. Data can be shared with qualified researchers in accordance with the conditions of ethical approvals and informed consents of the studies in compliance with GDPR and other relevant data protection regulations upon completion of a material transfer agreement with respective data controllers." I suggest rendering the data available without the signing of an MTA.

(Remarks on code availability)

Version 1:

Reviewer comments:

Reviewer #1

(Remarks to the Author)

The authors have satisfactorily addressed all my concerns.

(Remarks on code availability)

N.A.

Reviewer #2

(Remarks to the Author)

The authors have thoroughly addressed all my comments. I am pleased with the current state of the manuscript and have no further comments to add.

(Remarks on code availability)

Reviewer #3

(Remarks to the Author)

The authors have addressed the comments raised in the initial review.

(Remarks on code availability)

Reviewer comments in **black**
Comment responses in **red**
Manuscript extracts in **blue**

REVIEWER COMMENTS

Reviewer #1, expertise in prostate epithelial lineage hierarchy and club-like cells (Remarks to the Author):

Kiviaho A et al did an extensive spatial profiling analyses of prostate cancer cells and their associated microenvironment using 80 sections from 56 patients containing treatment naïve BPH and prostate cancer tissues, neoadjuvant treated prostate cancer and castration resistant prostate cancer specimens. The authors show that club-like cells display a senescence-associated secretory phenotype and there is an increased MDSC cells proximal to club-like cell regions in the tumor specimens. Some potential ligand-receptor interactions between club-like cells and MDSC are also revealed through the study. The study is well designed with proper controls. The results are described clearly. The spatial transcriptomics analysis is comprehensive and provides novel information regarding the cellular compositions and cell-cell interactions in human prostate specimens. The novel information obtained from this study will benefit the prostate cancer research community and help advance our understanding of the disease. Below are my major and minor comments.

Major concerns

- 1) The conclusion is exclusively based on the spatial transcriptomic analysis. It would be nice if the authors could corroborate at least some of the major conclusions using a different technology, such as multiplex RNA-in-situ or IHC analysis. For example, the higher expression of certain chemokines at club-like cell regions. I understand that it is difficult to identify PMN-MDSC using single markers, but at least using some relevant markers reported in this manuscript such as S100A9 etc. would make the conclusion more convincing.

We thank the reviewer for raising this point. To address these concerns, we carried out multiplex immunohistochemistry staining of 16 FFPE primary untreated prostate tumors from an independent cohort using antibodies specifically intended to detect club-like cells (PanCK, LTF, PIGR, CP), and PMN-MDSCs (CD45, CD66b, CD11b, CXCR2). In choosing the antibodies to stain for PMN-MDSCs, we relied upon Bronte et al. (PMID: 27381735) who have reported that "In human peripheral blood mononuclear cell (PBMC), the equivalent to PMN-MDSC are defined as CD11b⁺CD14⁻CD15⁺ or CD11b⁺CD14⁻CD66b⁺...". Additionally, Guo et al. (PMID: 37844613) state that "Prostate cancer is frequently infiltrated by myeloid inflammatory cells, including CD11b⁺HLA-DR^{lo}CD15⁺CD14⁻ cells (also termed polymorphonuclear myeloid-derived suppressor cells (PMN-MDSCs) or tumour-associated neutrophils)...". Guo et al. also show that the neutrophil chemokine receptor CXCR2 is nearly exclusively expressed in PMN-MDSCs.

Thus, by combining the immune cell marker *CD45* and granulocyte marker *CD66b* with *CD11b* and *CXCR2* we stain for PMN-MDSCs.

From the stained whole-slide images, we selected 101 approximately 3 mm² regions of interest (ROIs) that were either club-like negative (absent *LTF* staining, n = 54) or club-like positive (moderate to strong *LTF* staining, n = 47). We trained a random trees cell classifier to quantitatively assess the number of each cell type present in individual ROIs, hypothesizing that there would be more PMN-MDSCs in the club-like positive than in the club-like negative regions. We found that the club-like positive ROIs had a significantly higher proportion of PMN-MDSCs than the club-like negative ROIs ($p = 1.7 \times 10^{-8}$, two-sided Wilcoxon rank-sum test). Furthermore, the number of club-like cells in an ROI was positively correlated with the number of PMN-MDSCs ($\rho = 0.64$, $p = 3.6 \times 10^{-13}$), similar to what we reported based on the spatial transcriptomics analysis. We propose that the enrichment of *CD45*⁺*CD66b*⁺*CD11b*⁺*CXCR2*⁺ PMN-MDSCs in club-like positive regions corroborates our observation of proximity between club-like cells and PMN-MDSCs in these regions.

The results of this analysis are described in the manuscript:

*To validate the observed proximity between club-like cell prevalence and PMN-MDSC infiltration, we performed multiplex immunohistochemistry staining of 16 primary untreated prostate tumor tissue sections using antibodies against previously validated markers for these cell types (Methods). From the resulting staining images, we selected 101 approximately 3 mm² regions of interest that were either club-like negative (absent *LTF* staining, n = 54) or club-like positive (moderate to strong *LTF* staining, n = 47) (Figure 4e, Figure 4f, Supplementary Figure S7). We then trained a random trees cell classifier for seven cell categories, including PanCK+*LTF*+/*PIGR*+ club-like cells and *CD45*+*CD66b*+*CD11b*+*CXCR2*+ PMN-MDSCs (Figure 4g, Methods).*

Across all ROIs, the number of club-like-classified cells correlated positively with the number of PMN-MDSC-classified cells (Figure 4h). A higher proportion of PMN-MDSCs was present in club-like positive ROIs than in club-like negative ROIs ($p = 1.7 \times 10^{-8}$, two-sided Wilcoxon rank-sum test) (Figure 4i). No difference in the total number of detected cells between club-like positive and negative ROIs was observed ($p = 0.23$, two-sided Wilcoxon rank-sum test), while a greater number of club-like cells was present in the club-like positive ROIs ($p = 3.0 \times 10^{-15}$) (Supplementary Figure S7). Taken together these results demonstrate that the presence of club-like cells is strongly associated with PMN-MDSC infiltration and immunosuppressive activity in the prostate TME.

- 2) It would be interesting if the authors could check the expression profiles of club-like regions in different prostate cancer specimens and check if they can obtain additional information regarding how the expression profiles of the club-like cells are different in benign tissues versus tumor adjacent tissues and in treatment naïve tumors versus neoadjuvant-treated or castration resistant prostate cancer specimens.

We thank the reviewer for this suggestion. To address this, we carried out an additional differential gene expression analysis comparing the normalized expression of the Club region across samples in different treatment categories. We defined differentially expressed genes (DEGs) as having an absolute \log_2 fold-change above 1, and below 0.05 two-sided Wilcoxon rank-sum test p-value adjusted for multiple testing.

The results from this analysis should be interpreted with caution. Since SCM regions are defined by the underlying gene expression, DEGs are confounded by this initial categorization. Furthermore, effective normalization is a prerequisite for cross-sample comparison of expression, and the inherent heterogeneity in spatial transcriptomics data (varying numbers of cells and cell types in each data point, success in tissue permeabilization, sequencing depth) makes normalization challenging. We feel, however, that setting strict criteria for DEGs provides some confidence in the analysis.

The re-definition of these criteria allowed us to specify the statements made on lines 242-250. Here we reported an increase in expression of *WFDC2*, *KRT19*, *KLF5*, *ATP1B1*, *KRT4*, and *CCL20* post-treatment, results which were based on a pseudo-bulked region expression analysis with only a Wilcoxon rank-sum test p-value criteria. We also generated an additional plot (Supplementary Figure S3) and table (**Supplementary Table S6**) of the results. The analysis results are described in the manuscript:

“To test how treatment affected gene expression in the Club region specifically, we calculated differentially expressed genes between the Club regions of BPH, TRNA, NEADT, and CRPC sample categories (Supplementary Table S6). The expression of AR-regulated genes ABCC4, KLK3, MAF, NKX3-1, and PMEPA1 was lower in NEADT than TRNA samples (\log_2 fold-change ≤ -1 , $p_{adj} < 0.05$, two-sided Wilcoxon rank-sum test). The expression of canonical club cell marker LTF was likewise downregulated, while the expression of MMP7, PIGR, SCGB1A1, and SCGB3A1 was unaffected. Mouse luminal progenitor marker genes³⁶ S100A11, WFDC2, KRT19, KLF5, ATP1B1, KRT4, and MET were region-specific markers for the Club region, and their expression was similarly unperturbed by treatment.

Compared to BPH, NEADT, and CRPC, the most significantly upregulated DEGs in the TRNA Club region included AR-regulated luminal cell markers such as KLK3, KLK2, PMEPA1, MSMB, and ACP (Supplementary Figure S3). Club-like cells in tumor tissue have been reported to have increased AR-signaling activity compared to healthy prostate tissue¹⁷. The upregulation of activating transcription factors FOS and JUN was also detected in TRNA samples when compared to BPH samples, the expression of which has been previously linked to epithelial cell stress response in tumor progression¹⁷.”

Minor concerns

- 1) In the abstract the authors stated that the data were from 110 patients. In the first paragraph of the result section, the authors mentioned 56 patients. But the Method section only mentioned 45 patients.

We thank the reviewer for allowing us to clarify these statements. Altogether, we analyzed transcriptomics data from a total of 110 patients. This included publicly available scRNA-seq data from 64 patients (**Supplementary Table S2**, Sheet 2) and in-house generated ST data from 56 patients (**Supplementary Table S1**). The ST data was generated in three locations, Tampere University (37 patients, 37 samples), KU Leuven (11 patients, 11 samples), and NTNU Trondheim (8 patients, 32 samples). Given that both the Tampere and Leuven datasets contained neoadjuvant-treated samples, we combined the two to form the discovery cohort (48 samples), while the NTNU dataset (32 samples) was used as a standalone validation cohort. We have modified the text on to better describe this:

For the discovery cohort, a total of 48 prostatectomy samples were collected at Tampere University Hospital (37) and UZ Leuven (11).

- 2) The other marker for the club cell is PSCA. How the expression of PSCA is in the datasets?

We thank the reviewer for bringing up this point. In our data, *PSCA* is a marker for the Luminal region with $p_{\text{adj}} = 9.6 \times 10^{-6}$ (**Supplementary Table S5**), but not a Club region marker with $p_{\text{adj}} = 0.81$. Based on these results we conclude that while *PSCA* is an epithelial cell lineage marker, it is not necessarily expressed together with *MMP7*, *PIGR*, *CP*, and *LTF* that were used to annotate the Club regions.

- 3) Line 434, the authors should consider citing PMID: 26418304 when describing the presence of club cells in proximal region of mouse prostatic ducts.

We thank the reviewer for pointing to this article and have cited it accordingly.

Reviewer #2, expertise in scRNA-seq and spatial transcriptomics (Remarks to the Author):

Overview:

The manuscript presents a comprehensive cohort study that includes 10x Visium profiles of 80 human prostate tissues from 56 individual patients. To enhance the spatial data references, the authors integrated scRNA-seq data from seven previous studies, which included both normal and tumor prostate tissue samples. Using Non-negative Matrix Factorization (NMF), eight single-cell mapping-based (SCM) regions were defined. All biological insights were derived at the level of these SCM regions. Overall, the manuscript is well-written and easy to follow. The study's large sample size ensures the validity of the biological results. The bioinformatics analyses were conducted properly, with detailed information on the software tools used, including their versions and functions.

Major comments:

- 1) The most significant results of this manuscript are based on the eight single-cell mapping-based (SCM) regions defined using comprehensive scRNA-seq references. To determine the optimal number of SCM regions, the authors performed iterative Non-negative Matrix Factorization (NMF) with different numbers of components, ranging from 5 to 12. They selected the smallest number of components where no single cell state was the highest contributor in multiple components, as illustrated in Figure S9. However, according to Figure S9, no single cell state was the highest contributor in multiple components when the number of components was 5, 6, or 7. It is unclear why 8 components were ultimately chosen. Given the frequent mention of these eight SCM regions throughout the manuscript, it is important to clarify the rationale behind this decision.

We thank the reviewer for pointing this out and apologize for the confusion. This is a mistake in the text which should have – and has now been modified to – read as follows

*“The iteration with the **highest** number of components where no single cell state was the highest contributor in multiple components...”*

The reasoning for this approach was to find a number of components (regions) that reflected the transcriptional heterogeneity across our dataset while avoiding redundancy in the form of regions with highly similar cell-state contributions. For example, NMF iteration with $k = 9$ yields two regions (components 2 and 5) where the *muscle*-annotated cells have the highest component weight. In our view, this rule-based approach is unambiguous in that it doesn't allow for *post hoc* modification of the number of regions.

- 2) This is a follow-up to the previous point. Two of the eight SCM regions defined from the reference scRNA-seq datasets are muscle and fibroblast. However, as noted by the authors in Figure S12, "fibroblast and muscle categories were merged into one due to their similarity in terms of marker gene expression." If these two cell types are so similar, why do they

appear as distinct SCM regions after NMF? Should these two SCM regions be considered as one? Some explanation and justification is needed here.

We thank the reviewer for pointing out this apparent inconsistency. In analyzing the scRNA-seq data we chose a hierarchical approach of 1) assigning cells into broad cell-type categories and 2) finding heterogeneous expression within each cell-type category. While we weren't confident in separating these two cell types in step 1, they were organically separated in the data-driven analysis performed in step 2.

More specifically, we used the same cell-type gene markers reported in the original publications to create the broad cell-type categorization. When inspecting the dotplot presented in Supplementary Figure S13, we found that VI clusters 8 and 14 could not be unequivocally annotated as fibroblasts or muscle (i.e. myocytes). We were cautious about making this distinction especially since a subset of cancer-associated fibroblasts are known to adopt a muscle-like contractile phenotype and express some of the genes associated with myocytes (Sahai *et al.*, 2020, PMID: 31980749). Instead, we opted to collectively annotate clusters 8 and 14 as fibroblast/muscle and subjected them to the NMF-based algorithmic analysis to identify gene expression modules (Supplementary Figure S14). This analysis resulted in four unique modules, including standalone modules for fibroblast (*SERPINF1*, *DCN*, *LUM*, ...) and muscle (*MYL9*, *TPM2*, *DSTN*, ...) (**Supplementary Table S2**). These gene modules were then used to create the single-cell reference that was mapped onto the ST data, ultimately resulting in the *Fibroblast* and *Muscle* SCM regions. We have modified the referenced text excerpt to state the following:

“Fibroblasts and muscle cells could not be unequivocally separated based on marker gene expression and were thus included in the same cluster for downstream analysis. This procedure resulted in 10 unique cell-type clusters (Supplementary Figure S13d).”

Minor comment:

- 1) The integration analysis of all the scRNA-seq reference datasets (98 samples from 64 patients) is crucial, as it forms the basis for defining the 8 SCM regions. However, all related results and figures are only found in the supplementary material. Including some of these in the main manuscript would make more sense.

We thank the reviewer for this suggestion. We agree that the scRNA-seq analysis is a crucial part of the overall approach. However, we are cautious of the possible confusion arising from multiple cell-state categorizations presented in the main text. This is why we consciously chose to present the scRNA-seq data analysis entirely in the supplementary material. Following this suggestion, however, we have included a broad cell-type labeling legend in **Figure 1a** to provide more context in the main text. We hope this proves useful to the reader.

Reviewer #3, expertise in myeloid-derived suppressor cells and the prostate cancer TME
(Remarks to the Author):

In the present manuscript, Kiviaho et al. highlight the role of club-like epithelial cells in the tumor microenvironment of prostate cancer. They propose that this subset of epithelial cells may be linked to the presence of tumor-infiltrating PMN-MDSCs. The authors utilized single-cell RNA-sequencing and spatial transcriptomics from prostate cancer samples of 110 patients. Overall, the manuscript is technically solid and interesting. The strength of this work lies in the spatial transcriptomics data generated from valuable patient samples. Previous studies, such as Henry et al. (Cell Reports 2018), have reported cells with the same signature as club cells in healthy and cancerous prostate tissue. This manuscript adds to the current knowledge by showing that these cells express elevated levels of canonical neutrophil chemokines and are associated with increased PMN-MDSC activity in primary and metastatic tumors, an immune population already demonstrated to have pro-tumorigenic roles in CRPCs. Although the manuscript is intriguing and the data are unique, there are some minor concerns to address:

- 1) The authors do not provide a clear cause-and-effect experiment to support the statement in the abstract that "club-like cells induce myeloid inflammation." It is suggested to revise this statement.

We thank the reviewer for raising this point and have revised the statement in question:

"Our results indicate that club-like cells are associated with myeloid inflammation previously associated with androgen deprivation therapy resistance, providing a rationale for their therapeutic targeting."

- 2) The manuscript indicates that the presence of various chemokines is a sign of SASP (line 407). However, the molecules found may be involved in too many mechanisms to be defined as SASP.

We thank the reviewer for pointing to this lack of rigor in determining whether the Club region has an active senescence-associated secretory profile (SASP). To address this more methodologically, we identified a signature of genes that encode proteins whose secretion is increased in SASP, as indicated in Table 1 of Coppé et al. 2010 (PMID: 20078217). We named this signature of 58 genes "SASP upregulated" and included it among the signatures used in the study (**Supplementary Table S7**). We found that genes in the SASP upregulated signature were overrepresented among the Club region markers ($p_{\text{adj}} = 7.2 \times 10^{-4}$, one-sided Fisher's exact test, **Supplementary Table S8**). Altogether ten genes from this signature were region-specific markers in the Club region: *CCL20*, *CTSB*, *CXCL1*, *CXCL2*, *CXCL3*, *CXCL8*, *ICAM1*, *IGFBP3*, *MMP10*, and *PLAU*.

To present these results we have added the following statement to the manuscript:

“Club region-specific markers were enriched for genes that encode for proteins whose secretion is increased in the senescence-associated secretory phenotype⁵¹ (SASP), suggesting that club-like cells engage in similar secretory activity ($p_{adj} = 7.2 \times 10^{-4}$).”

We have further discussed the result in the *Discussion*-section of the manuscript:

“Here we report that club-like cells have upregulated expression of genes that encode for proteins whose secretion is increased in SASP⁵¹, including TME-altering proteases cathepsin β (CTSB), stromelysin-2 (MMP10), and urokinase (PLAU), insulin-like growth factor-binding protein 3 (IGFBP3), and intercellular adhesion molecule 1 (ICAM1). Furthermore, club-like cells display elevated expression of similarly SASP-associated myeloid cell chemotaxis-inducing chemokines^{51,53} CXCL1, CXCL2, CXCL3, CXCL8, and CCL20...”

We hope this additional analysis – alongside the previously presented results – is sufficient to justify the statement on the club-like cells having a SASP-like expression profile.

- 3) The conclusions are very definitive and should be softened. For example, the sentence on line 254, where the presence of transcripts for chemokines involved in neutrophil recruitment, is not proof of active neutrophil recruitment into the tumor.

We are grateful to the reviewer for addressing this frailty. We have revised this sentence to state the following:

*“While the sensitivity of ST was not sufficient to detect significant levels of neutrophil chemotactic receptor CXCR2 expression, the elevated expression of its canonical ligands CXCL1, CXCL2, and CXCL8 **suggests that club-like cells partake in their recruitment to the TME.**”*

We have also modified the statement made in the last paragraph of the discussion:

*“Taken together, our analysis highlights club-like cells as a key TME constituent that **is associated with PMN-MDSC infiltration.**”*

- 4) Given that the strength of this work lies in the methodology used and the data generated, these should be made available to the scientific community. Currently, the reporting summary states: "The data that support the findings of this study are available from the authors but restrictions apply to the availability of these data. Data can be shared with qualified researchers in accordance with the conditions of ethical approvals and informed consents of the studies in compliance with GDPR and other relevant data protection regulations upon completion of a material transfer agreement with respective data controllers." I suggest rendering the data available without the signing of an MTA.

We thank the reviewer for this suggestion. We have deposited the gene counts and imaging data of the discovery set and the metastatic samples onto Gene Expression Omnibus under the accession code **GSE278936**. The following secure token has been created to allow review of record **GSE278936** while it remains in private status: **khwtgwmmrtqjjuf**. Raw sequencing data from the validation cohort has been deposited onto the European Genome-Phenome Archive under access code **EGAD5000000603**. This is the maximal level of openness achievable under the current regulations.

Reviewer comments, 2nd revision round

Reviewer #1 (Remarks to the Author):

The authors have satisfactorily addressed all my concerns.

Reviewer #1 (Remarks on code availability):

N.A.

We thank the reviewer for their contribution to the review process.

Reviewer #2 (Remarks to the Author):

The authors have thoroughly addressed all my comments. I am pleased with the current state of the manuscript and have no further comments to add.

We thank the reviewer for their feedback on the manuscript.

Reviewer #3 (Remarks to the Author):

The authors have addressed the comments raised in the initial review.

We thank the reviewer for providing comments on our work.